

# Detection of unsafe workplace behaviors: Sec-YOLO model with FEHA attention

Yang Liu, Shuaixian Liu, Jie Gao, Tao Song and Wenyu Dong

School of Electronic and Information Engineering, Hebei University of Technology, Tianjin, China

## ABSTRACT

Detecting unsafe human behaviors is crucial for enhancing safety in industrial production environments. Current models face limitations in multi-scale target detection within such settings. This study introduces a novel model, Sec-YOLO, which is specifically designed for detecting unsafe behaviors. Firstly, the model incorporates a receptive-field attention convolution (RFAConv) module to better focus on the key features of unsafe behaviors. Secondly, a deformable convolution network v2 (DCNv2) is integrated into the C2f module to enhance the model's adaptability to the continually changing feature structures of unsafe behaviors. Additionally, inspired by the multi-branch auxiliary feature pyramid network (MAFPN) structure, the neck architecture of the model has been restructured. Importantly, to improve feature extraction and fusion, feature-enhanced hybrid attention (FEHA) is introduced and integrated with DCNv2 and MAFPN. Experimental results demonstrate that Sec-YOLO achieves a mean average precision (mAP) at 0.5 of 92.6% and mAP at 0.5:0.95 of 63.6% on a custom dataset comprising four common unsafe behaviors: falling, sleeping at the post, using mobile phones, and not wearing safety helmets. These results represent a 2.0% and 2.5% improvement over the YOLOv8n model. Sec-YOLO exhibits excellent performance in practical applications, focusing more precisely on feature handling and detection.

## INTRODUCTION

Throughout the entire process of corporate development, the foundation of safety in production is invariably its cornerstone. Despite significant improvements in industrial safety management in recent years, the risk of production accidents remains substantial. Human factors, which account for the majority of incidents, are particularly responsible. This indicates that existing safety management measures have yet to effectively address the safety challenges posed by increasingly complex and diverse working environments. Indeed, human factors play a decisive role in ensuring safety in production (*Wenwen et al., 2011*). Numerous studies have shown that most industrial accidents are caused by improper operations, including a lack of safety awareness and lax attitudes towards work (*Xie & Guo, 2018*). For instance, an analysis of accidents in the Greek petrochemical industry from 1997 to 2003 revealed that 73% of the accidents could be attributed to human (46%) and organizational (37%) factors (*Konstandinidou et al., 2006*). Similarly, an

Corresponding author
Tao Song, songtao@hebut.edu.cn

analysis of severe industrial accidents caused by hazardous chemicals in South Korea from January 2008 to June 2018 indicated that approximately 76.1% of chemical incidents were due to human error. Although the causes of accidents varied, the recurrent nature of similar types of chemical incidents suggests systemic issues. These incidents were primarily caused by operational errors, which also highlight deficiencies in the implementation and dissemination of safety measures (*Jung, Woo & Kang, 2020*). Therefore, human factors undoubtedly represent a critical aspect of industrial safety.

Although strategies such as safety education, safety criticism, behavioral monitoring, and feedback provision are widely employed in safety management, they often fail to ensure that workers fully understand the severity of their actions. As a result, they prove ineffective in ensuring long-term safety. Furthermore, behavioral modification largely depends on the specific factors addressed during safety education, which presents challenges in ensuring that workers fully comprehend and reflect on their errors (*Yang et al., 2023*; *Morgan et al., 2021*; *Gonyora & Ventura-Medina, 2024*). Meanwhile, intelligent electronic surveillance methods based on target detection can instantly alert workers at the moment an incorrect behavior occurs, or even as an erroneous trend emerges, effectively preventing potential safety risks and production accidents. Currently, in many work environments requiring special monitoring, the application of intelligent electronic surveillance is becoming increasingly widespread, primarily used to detect abnormal human behaviors and other unsafe factors. Nevertheless, current intelligent monitoring methods still face a series of challenges, such as high target density, scene occlusion, and multi-scale detection issues. This is especially true in complex industrial environments, where traditional deep learning models often fall short in terms of detection accuracy and real-time performance.

Detecting unsafe behaviors in industrial environments presents a variety of challenges. One of the most common issues encountered when using computer vision techniques is visual occlusion. When workers are partially or fully blocked by objects, most vision-based methods fail to detect and monitor their actions. Additionally, self-occlusion occurs when a worker's limbs or joints are obstructed by their own body—especially when facing away from the camera—making pose estimation and action recognition more difficult. Moreover, certain unsafe postures cannot be accurately recognized or classified, particularly in cases involving ambiguous or transitional movements. Although modern object detection methods can effectively recognize many objects in industrial scenes with high precision, their performance significantly deteriorates when dealing with small-scale objects or irregular postures. Furthermore, in industrial settings, target behaviors often involve frequent changes in body orientation and non-uniform motion, which poses additional challenges for robust detection. These complexities highlight the need for more specialized and optimized deep learning models capable of handling such variability in real-world conditions (*Liu et al., 2021*). In 2017, *Fang et al. (2018)* aimed to accurately identify safety helmets in distant scenes using the faster region-convolutional neural network (Faster R-CNN) model. They collected a substantial dataset of workers wearing safety helmets in various scenarios. The results demonstrated that this method improved monitoring detection precision and speed across different settings. However, the detection

speed was too slow to meet the requirements for real-time detection, and its performance in complex scenarios remained untested. In 2021, *Kong et al. (2021)* combined computer vision with long short-term memory networks (LSTMs) to predict unsafe human behaviors. Initially, they used SiamMask to track the individuals to be detected, followed by an improved Social-LSTM to predict future human behavior trajectories. Finally, the point inclusion in polygon (PNPOLY) algorithm was employed to determine the presence of unsafe behavior (*Kong et al., 2021*). Although combining various algorithms and models yielded better results, the target detection area required manual segmentation. It was also limited to construction site scenarios, which made it non-generalizable. Similarly, in 2023, *Park et al. (2023)* proposed that deep learning networks are an effective means to replace safety managers in monitoring and managing workers. They reconstructed the YOLOv5 model with an attention mechanism and optimized loss function, focusing on ladder workers and marking the hinge parts as target labels. They estimated the actual working height of the object detection bounding box, using it as the dataset, thereby enhancing the detection network's recognition accuracy (*Park et al., 2023*). However, the network was still limited to specific scenarios and was not effective in generalizing to other scenes for detecting unsafe human behaviors. To address challenges such as object density, occlusion, and multi-scale scenes in classroom video images, *Chen, Zhou & Jiang (2023)* improved the YOLOv8 model by proposing a novel C2f_Res2block module and integrating multi-headed self-attention (MHSA) and efficient multi-scale attention (EMA) into the model. Although Res2block and MHSA can significantly improve model performance, they also introduce high computational costs. In particular, MHSA, as a self-attention mechanism, requires substantial computation and memory, making its use in real-time monitoring highly challenging. Despite recent advancements, existing detection models still face several limitations when applied to industrial unsafe behavior monitoring. Many suffer from poor generalization across different scenes, low recall rates under occlusion or complex postures, and high computational demands that hinder real-time deployment. These shortcomings significantly reduce their reliability in high-risk industrial settings.

To address these issues, we propose Sec-YOLO, a task-driven enhancement of YOLOv8n tailored for complex industrial scenarios. Sec-YOLO integrates three key improvements: (1) receptive-field attention convolution (RFAConv) to enhance spatial focus; (2) a DCNv2-FEHA module for adaptive geometric modeling of irregular human behaviors; and (3) a multi-branch auxiliary feature pyramid network (MAFPN) to strengthen multi-scale feature fusion.

At the core of Sec-YOLO lies the proposed feature-enhanced hybrid attention (FEHA) module, which combines global and local attention across spatial and channel dimensions. FEHA is uniquely embedded in both the deformable convolution offset path and deep fusion layers, allowing it to guide feature learning more precisely and dynamically adapt to complex unsafe behaviors. This novel attention design is the central innovation of our work and plays a critical role in achieving robust, real-time detection across diverse scenarios.

This article proposes a novel model, Sec-YOLO, for detecting unsafe human behaviors. To enhance the model's detection capability, an enhanced hybrid attention module, FEHA, is introduced. The specific contributions are as follows:

(a) The model employs RFAConv to replace the standard Conv in YOLOv8, thereby achieving a more comprehensive representation of receptive field spatial features.

(b) The introduction of the second-generation deformable convolution, DCNv2, into the C2f module enhances model detection capabilities. This module operates by calculating offset vectors to perform feature extraction from feature maps, particularly improving detection performance in scenarios involving irregular behaviors depicted in images.

(c) To enhance the multi-scale feature fusion capabilities, the neck part of the model employs the MAFPN architecture. This architecture comprises two key components: the superficial assisted fusion (SAF) module and the advanced assisted fusion (AAF) module. The SAF module is dedicated to preserving optimal shallow features, while the AAF module focuses on integrating diverse features to achieve more refined and extensive feature fusion.

(d) This study introduces the FEHA attention module, which is embedded within the second generation deformable convolution (DCNv2) to enhance the capability of capturing offsets during the convolution process. Additionally, the FEHA module is positioned after the convolution within the MAFPN architecture that is utilized for advanced feature fusion, thereby enhancing the effectiveness of feature integration. The operational mechanism of FEHA involves using convolution to augment pooled information, mixing the initial attention map with the original input, and subsequently extracting attention features to enrich the model's capacity for information aggregation. This enables the model to dynamically adjust the significance of each channel and spatial position within the feature map, thereby responding more adeptly to changes in the spatial content context.

(e) Conducting experimental tests on a dataset of unsafe behaviors collected in the network to demonstrate the effectiveness of the proposed Sec-YOLO.

The subsequent sections of the article are organized as follows: "Related Work" presents the methodology of the model design, "Methods" discusses the improvements made to the model, "Experiment and Analysis" details the experimental analysis results and discussions, and "Conclusions" concludes the work of this article.

## RELATED WORK

In this section, we first provide a detailed review of the target detection algorithm YOLOv8n that is to be improved. Next, we introduce the key modules involved in this algorithm. Additionally, we briefly describe the application and impact of classical attention mechanisms in the field of object detection, particularly how they enhance the model's feature representation capabilities.

## Architecture of YOLOv8n

YOLOv8 is the latest model iteration released by Ultralytics following the YOLOv5 series, specifically designed for high-speed and precise object detection tasks. This model utilizes an anchor-free detection approach and incorporates dynamic quantization technology algorithms, allowing it to directly predict the centers and aspect ratios of objects to determine their categories and positions. Such a method substantially optimizes the model's processing speed and accuracy. The YOLOv8 series includes five variants of different scales: YOLOv8n, YOLOv8s, YOLOv8m, YOLOv8l, and YOLOv8x, each tailored to meet diverse application requirements.

Among these, YOLOv8n is the lightest model in the series. It not only maintains an outstanding detection speed, particularly suited for real-time monitoring of unsafe behaviors, but also achieves high detection accuracy with minimal parameter size and computational demand. The structure of YOLOv8n is shown in Fig. 1. In its backbone network design, YOLOv8n inherits the cross-stage partial (CSP) architecture concept from YOLOv5. However, in this iteration, the traditional C3 module is replaced with the newer C2f module to further reduce the model's weight. Additionally, YOLOv8n innovatively eliminates the convolutional structures in the upsampling phase of path aggregation network—feature pyramid network (PAN-FPN) in the neck section and replaces them with C2f modules, thereby enhancing performance and reducing complexity.

## RFAConv module

Receptive-field attention convolution (RFAConv) is a fixed convolutional combination that emphasizes the importance of different features within a sliding receptive field and sorts spatial features of the receptive field (*Zhang et al., 2023*). The structural principle of RFAConv is shown in Fig. 2.

RFAConv introduces an efficient mechanism for enhancing convolutional neural networks by leveraging $3 \times 3$ group convolutions to quickly capture spatial features across the receptive field. This mechanism improves network performance by enabling feature cross-aggregation, where information from different spatial locations is effectively combined to better understand the underlying data.

To minimize computational overhead, RFAConv uses an efficient strategy for aggregating global information within each receptive field. Specifically, it first applies average pooling (AvgPool) to gather global feature information across the receptive field. This global information is then merged using $1 \times 1$ group convolutions, allowing for effective feature fusion without adding significant computational cost.

Next, a SoftMax function is applied to the fused features, which assigns importance to different features within the receptive field. This emphasizes the most relevant information, enabling the network to focus on the most critical spatial details. Finally, the feature maps from the two branches—one with the spatial features and one with the aggregated global information—are combined to form the final output feature map.

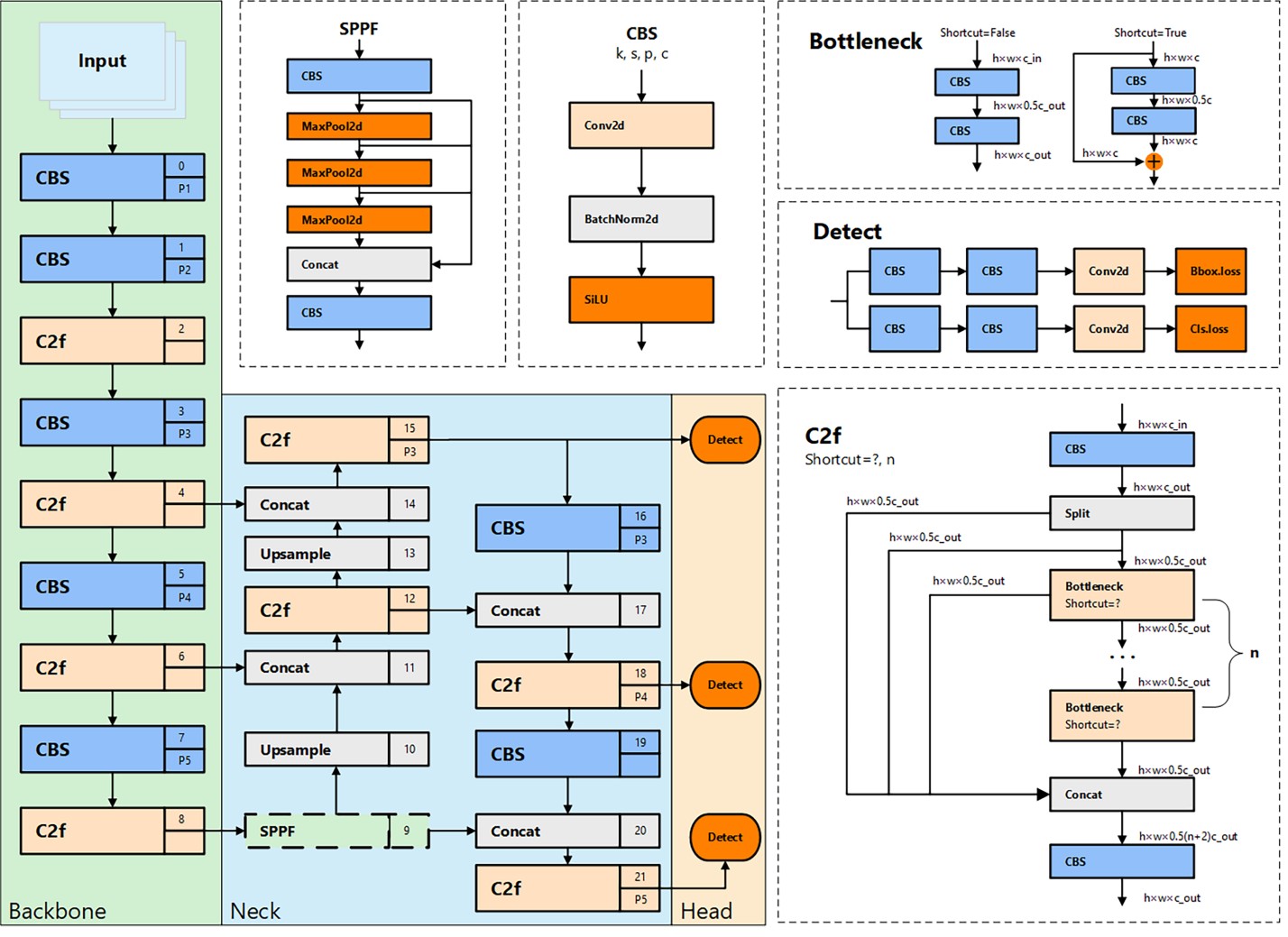

**Figure 1 The architecture and internal modules of YOLOv8.**

The overall computation process of RFAConv can be summarized as follows:

$$F = Softmax\left(g^{1\times1}(AvgPool(X))\right) \times ReLU\left(Norm\left(g^{1\times1}(X)\right)\right) = A_{rf} \times F_{rf}. \tag{1}$$

Here, $g^{1\times1}$ represents a group convolution of size i × i, k denotes the kernel size, Norm indicates normalization, and X denotes the input feature map. F is obtained by multiplying attention map $A_{rf}$ with transformed receptive field spatial features $F_{rf}$. The combination of SoftMax and ReLU is employed to enhance both spatial selectivity and activation sparsity. SoftMax ensures that the attention weights across the receptive field are normalized and interpretable as relative importance scores, while ReLU suppresses irrelevant activations by eliminating negative responses in Frf. This design enables the network to concentrate more effectively on salient receptive field regions while maintaining numerical stability during training. The structure follows the original RFAConv formulation.

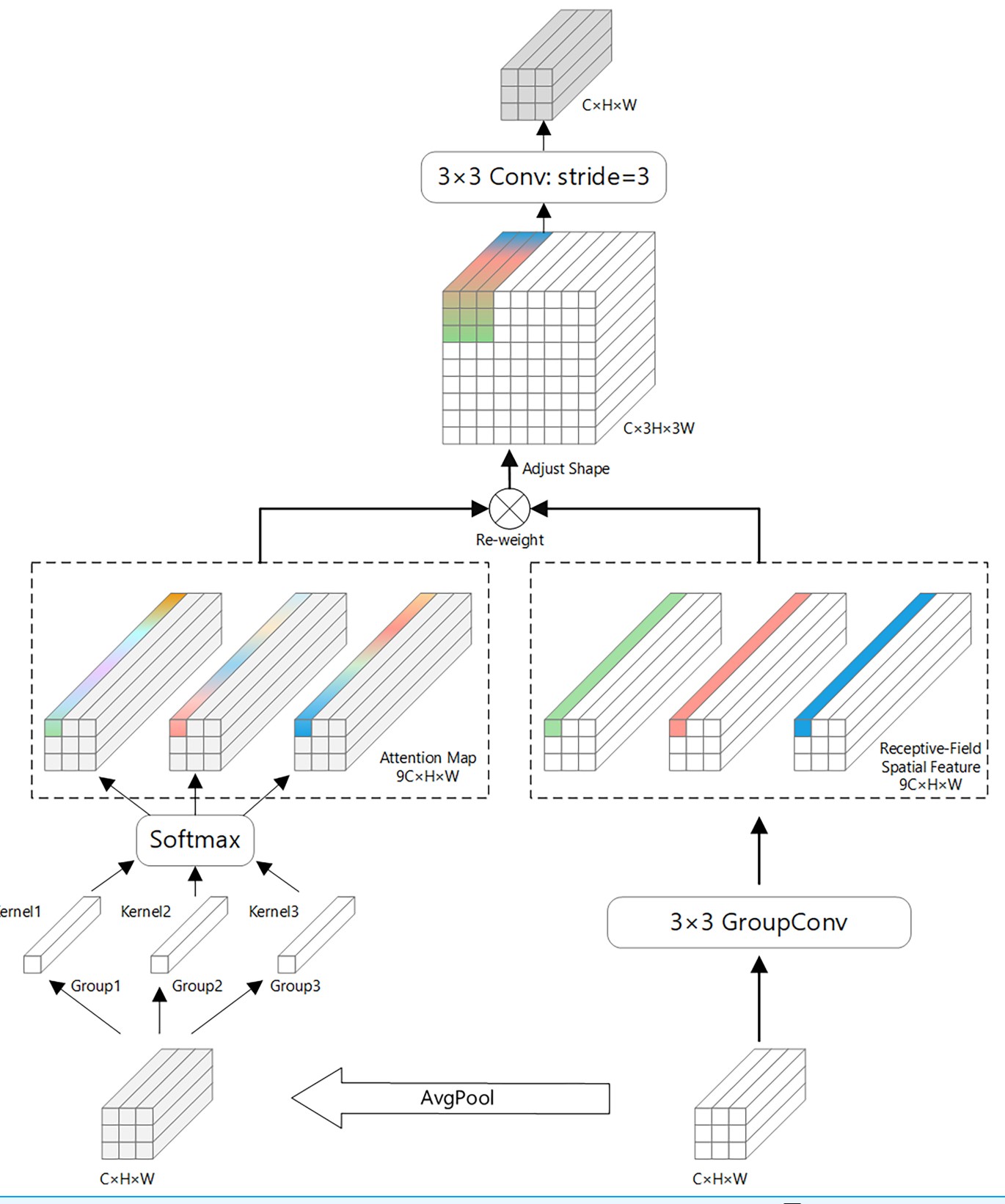

**Figure 2 Receptive-field attention convolution (RFAConv) structure diagram.**

## DCNv2 module

Deformable convolution network (DCN) enhances the capability of traditional convolutional neural networks to handle irregular geometric variations by introducing additional parameters (*Dai et al., 2017*). Initially, a batch of input images $U \in Rb \times h \times w \times c$ is processed by a conventional convolution layer with a kernel size $k$, producing an output $V \in Rb \times h \times w \times (2 \times k \times k)$, which has the same dimensions as $U$. Here, $V$ represents the offset at each pixel point in the original image, including offsets in both x and y directions. Subsequently, each pixel index in the original image $U$ is added to the offset computed in $V$ to calculate the new pixel positions after the offset. The pixel values at these new coordinates are then obtained using bilinear interpolation.

However, a major drawback of DCN is that the new positions of the sampling points, after applying the offset, may exceed the ideal sampling locations. This could lead to some DCN convolution points covering parts of the image irrelevant to the object's content. To further enhance DCN's ability to learn geometric transformations, an additional modulation mechanism is proposed to be incorporated into DCN. Besides learning the offset parameters $\Delta p$ (offset), a modulation parameter $\Delta m$ is introduced through modulation learning. This parameter $\Delta m$ helps to further reasonably control the range of the new sampling points. The improved version of DCN, commonly referred to as DCNv2 (*Zhu et al., 2019*), utilizes this additional modulation mechanism to enhance performance.

## MAFPN structure

The multi-branch auxiliary feature pyramid network (MAFPN) represents an innovative "neck" architecture, specifically designed to enhance the integration of shallow and deep features, thereby optimizing recognition capabilities across multiple scales. The MAFPN architecture comprises two key connectivity modules: superficial assisted fusion (SAF) and advanced assisted fusion (AAF) (*Yang et al., 2024*). As illustrated in Fig. 3, the implementation mechanisms of SAF and AAF are detailed. SAF is primarily applied to the superficial layers of the neck, aiming to retain shallow feature information from the backbone network to improve the detection efficiency of small-scale targets. Conversely, AAF is applied to the deeper layers, integrating features from the current and adjacent layers of varying resolutions, thereby significantly enhancing the model's target detection performance across various scales. This innovative configuration of the structure markedly improves the system's multi-scale target detection capabilities.

In Fig. 3, the subscript n in the lower right corner represents the baseline feature layer, with Pn − 1 and Pn + 1 respectively denoting the high-resolution features of the previous layer and the low-resolution features of the next layer. In the upper right corner, the apostrophe, as in P′n, indicates features obtained after SAF processing; double apostrophes, as in P″n, denote features obtained after AAF processing.

## Attention mechanism

Over the past several decades, computer vision has become one of the core research directions within the field of artificial intelligence. With the rapid development of deep learning, the performance of computer vision models has significantly improved.

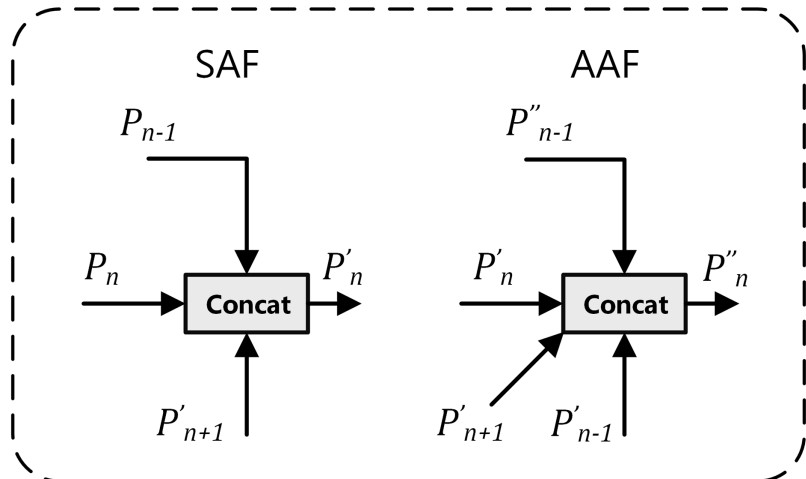

**Figure 3** Diagram: superficial assisted fusion (SAF) on the left and advanced assisted fusion (AAF) on the right.

However, the increasing complexity of the three major tasks, such as image classification, object detection, and semantic segmentation, has overwhelmed traditional feature extraction and classification methods when dealing with large-scale visual datasets and complex scenes. To address these issues, researchers developed the attention mechanism, inspired by the focused attention of the human visual system. This mechanism allows computer vision models to dynamically prioritize the most relevant regions of the images, significantly improving performance in tasks such as object detection and image segmentation.

A convolutional block attention module (CBAM) sequentially extracts attention features along two independent dimensions: channel and spatial dimensions (*Woo et al., 2018*). The primary idea of CBAM is straightforward, simultaneously considering channel attention (CAM) and spatial attention (SAM). Through multiple experiments, it has been determined that concatenating these two parts using residual structures achieves the best performance.

CBAM employs both global average pooling and global max pooling to transform the feature maps from dimensions $C \times H \times W$ to two different $C \times 1 \times 1$ spatial context information feature vectors. It uses shared multi-layer perceptron (MLP) and Sigmoid activation functions to generate channel feature attention maps, thereby capturing the performance of different features.

SAM takes the feature maps generated by CAM as inputs, processes them separately through global average pooling and global max pooling based on the channel dimension, obtaining two $1 \times H \times W$ feature vectors. After concatenating along the channel and passing through a $7 \times 7$ convolution kernel to extract features and reduce dimensionality, it generates spatial feature attention maps through Sigmoid activation. Finally, it reconnects with the input of the spatial attention part through residual connections to restore to the size of $C \times H \times W$, thus obtaining the final output.

Coordinate attention is implemented in two main steps (*Hou, Zhou & Feng, 2021*): coordinate information embedding and coordinate attention generation. In the coordinate information embedding step, the model aggregates features using average pooling along two directions to capture important spatial information. Subsequently, in the coordinate attention generation step, these features aggregated along different directions are concatenated and processed through convolutional layers and non-linear activation functions to generate intermediate feature maps. These intermediate feature maps are then further split into two independent tensors and transformed into feature vectors with the same number of channels as the input feature map using two separate $1 \times 1$ convolutions.

Finally, these two feature vectors along different spatial dimensions are processed through a sigmoid activation function and multiplied with the original input feature map, forming a residual connection. This structure not only strengthens the model's attention to spatial locations but also significantly enhances the model's performance in handling complex visual tasks through this complementary approach. Coordinate attention, with its unique structural design and processing flow, demonstrates an effective strategy for integrating precise positional information into modern deep learning models.

MLCA primarily employs three techniques for feature mapping and recovery to adapt to different information processing needs: local average pooling (LAP), global average pooling (GAP), and reverse average pooling (UNAP) (*Wan et al., 2023*). LAP focuses on extracting more refined local feature information by dividing the feature map into multiple $k \times k$ patches, performing global average pooling within each patch, and using an appropriate number of adaptive average pooling outputs to maintain the spatial continuity of the information. GAP utilizes adaptive average pooling to reduce the feature map to a $1 \times 1$ dimension, which aids in capturing overall or global feature information and effectively summarizing the extensive features of the entire image. Reverse average pooling (UNAP) is used to restore feature maps compressed by GAP or LAP back to their original size. This process is achieved by distributing stored weights to the corresponding patches, ensuring that the spatial dimensions of the feature map remain consistent with those at the input stage.

## METHODS

In this section, we introduce a newly proposed attention mechanism in this research, the feature-enhanced hybrid attention module (FEHA). Additionally, we present a novel algorithm, Sec-YOLO, which is based on the improved module and the FEHA attention mechanism, specifically designed for the detection of unsafe behaviors. The implementations of the FEHA attention mechanism and the Sec-YOLO model are publicly available at DOI: 10.5281/zenodo.14233625 to ensure reproducibility and facilitate future research.

### FEHA attention mechanism

Attention mechanisms have significantly enhanced the performance of neural networks by capturing and combing features from both channel and spatial dimensions. Although CBAM considers both channel and spatial attention mechanisms, the use of multi-layer perceptron increases the model complexity. Derived on CBAM, Hybrid attention module

(HAM) enhances feature extraction by utilizing both average pooling and max pooling in the channel attention mechanism (*Li et al., 2022*). It employs two learnable parameters to fuse the captured feature vectors, preserving original feature information while adaptively adjusting feature biases. In spatial attention, the channel separation technique multiplies the features obtained from channel attention, denoted as F′, with two predefined mask groups, resulting in two separate feature sets, F1′ and F2′. These are then subjected to two types of pooling along the spatial dimension, and the pooling results are concatenated to produce two sets of output features. Although this method effectively identifies salient features, it focuses solely on the spatial dimension and overlooks the positional information formed by the length and width dimensions. In ASF-YOLO, the channel and position attention mechanism (CPAM) attention fusion mechanism, similar to CBAM, combines both channel and spatial attentions (*Kang et al., 2024*). While it effectively captures positional information of features through pooling across the length and width dimensions in spatial attention, the exclusive use of average pooling in channel attention may result in the loss of crucial feature information.

To further enhance the performance of the attention mechanism, we propose the FEHA, which leverages the strengths of the existing methods. This module can comprehensively capture image features, demonstrating higher accuracy and generalization. Specifically, the structure of the FEHA attention module, as depicted in the Fig. 4, includes both a channel attention module and a positional attention module. This design effectively integrates the strengths of both attention types to optimize the extraction and utilization of feature information across different dimensions, significantly improving the performance of the system in complex visual environments.

Channel attention module: Fig. 5 shows the channel attention module of FEHA. The roles of max pooling and average pooling in different stages of image feature extraction have been well-documented. Max pooling may overlook subtle features, while average pooling might not effectively highlight significant features. Directly merging these two pooling methods could potentially lead to information loss. Therefore, to better integrate the feature information derived from both pooling methods and enhance the module's feature representation capability, We are introducing an additional branch as a compensatory mechanism to support and enhance the existing structure. This branch uses 2D convolution to preliminarily extract features from the input feature tensor X. Subsequently, average pooling aggregates these features into a vector of the same size as those from the other two branches. Both max pooling and average pooling utilized here employ local pooling with a size of 2, which helps preserve information to a greater extent. The outputs from the three branches are not simply added together; instead, we introduce learnable parameter $\alpha$ to facilitate feature selection. As illustrated in the Fig. 5, the results from all three branches are multiplied by respective learned parameters before being summed. This process can be formulated as follows:

$$X_c^{add} = (1 - \alpha) \otimes \left( X_c^{LAP} \oplus X_c^{LMP} \right) \oplus \alpha \otimes X_c^{EN}. \tag{2}$$

Here, $X_c^{add}$ represents the final output of the channel attention module, while $X_c^{LAP}$ and $X_c^{LMP}$ denote the outputs of input X after applying average pooling and max pooling,

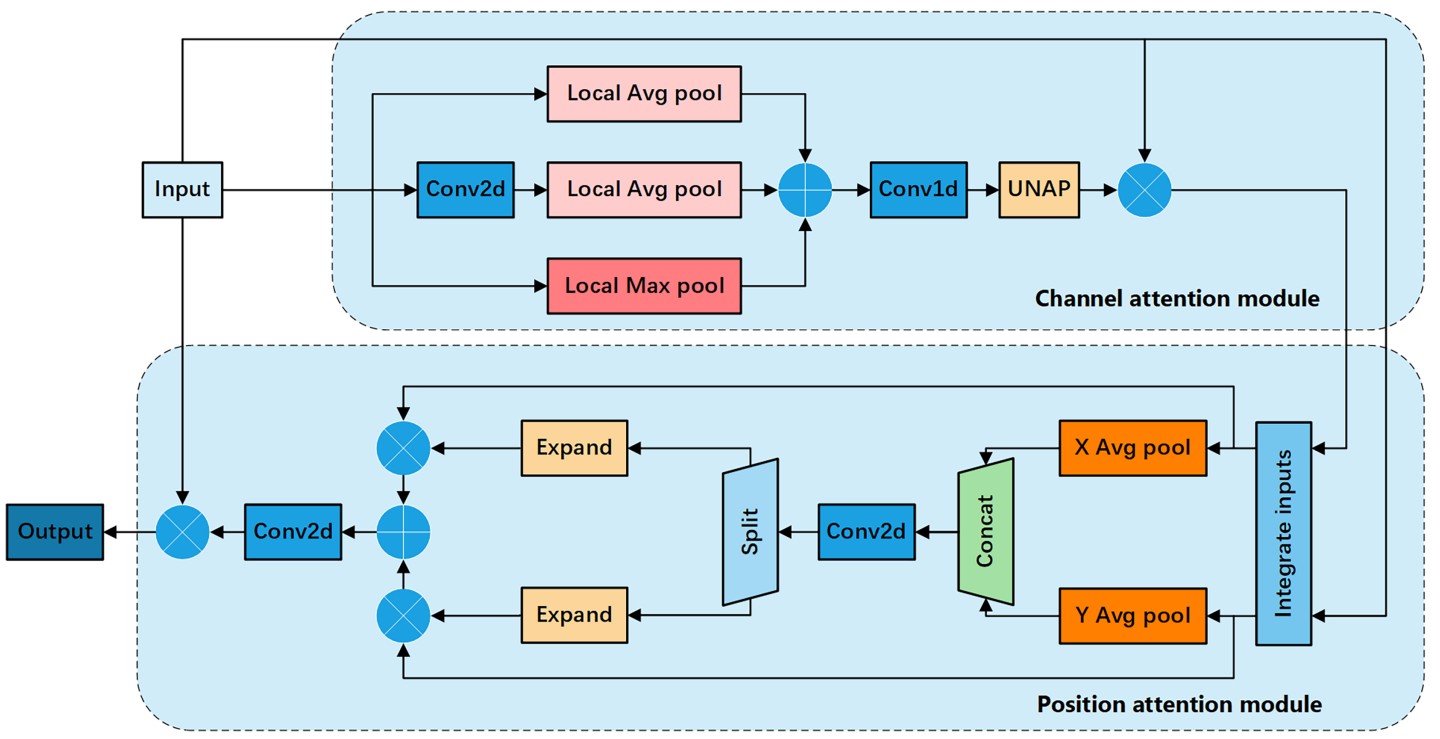

**Figure 4** Basic structure diagram of the feature-enhanced hybrid attention module (FEHA).

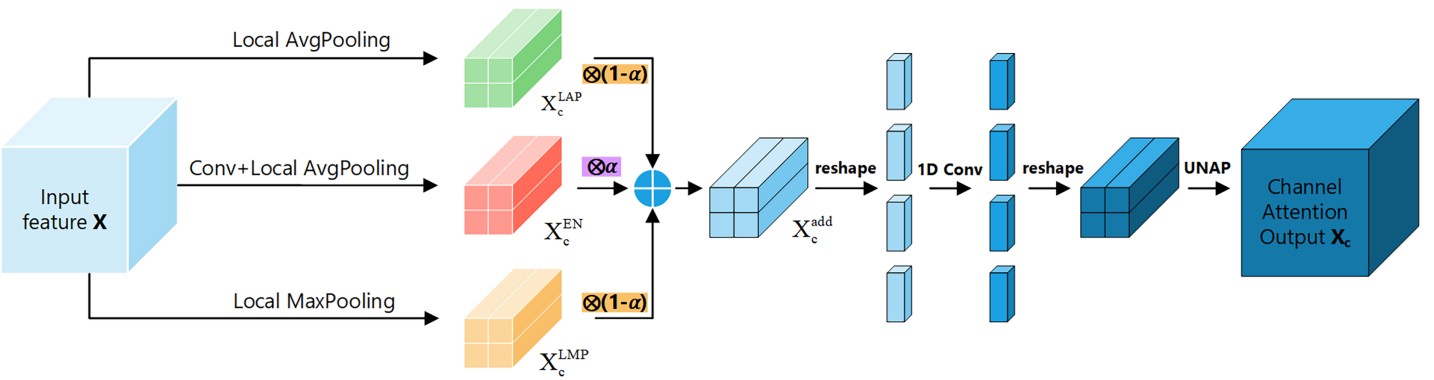

**Figure 5** Channel attention section of the feature-enhanced hybrid attention module (FEHA).

respectively. Additionally, $X_c^{EN}$ refers to the output of the enhancement compensation branch. The parameter $\alpha \in [0, 1)$ is a learnable scalar that balances the contribution between the combined pooling result and the enhancement branch. This formulation allows the model to adaptively adjust feature emphasis based on training data, enhancing representation capacity and robustness in channel attention learning. In our implementation, the initial value of $\alpha$ is set to 0.5, which was determined based on empirical evaluation. A detailed ablation analysis on $\alpha$ is presented in Section Hyperparameter selection.

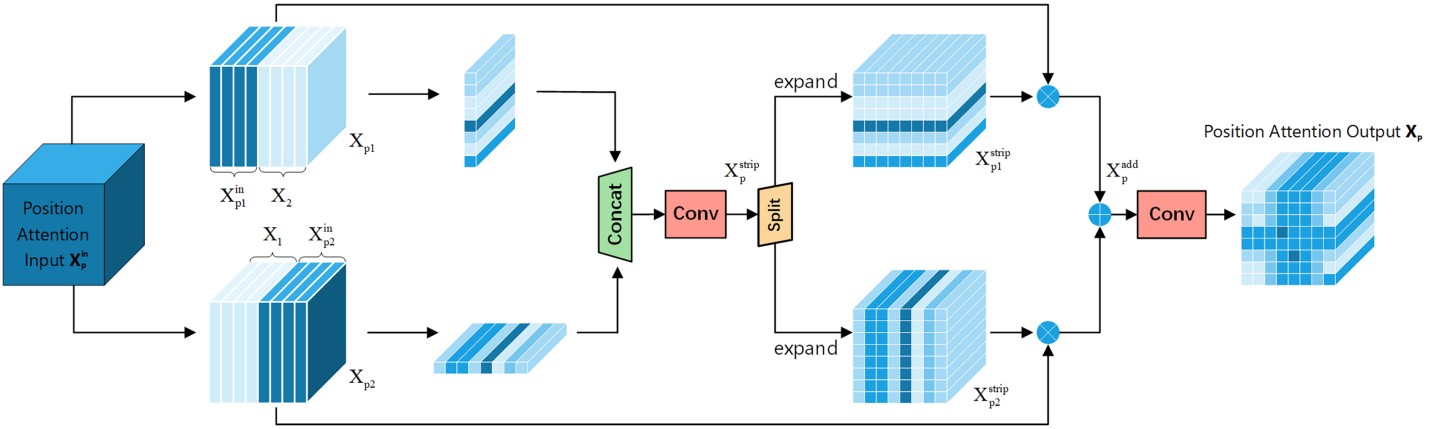

**Figure 6 Positional attention section of the feature-enhanced hybrid attention module (FEHA).**

After transforming X into $X_c^{add}$, reshape it along the channel dimension to obtain a new tensor of size $(C \times 2 \times 2) \times 1 \times 1$. Then, use 1D convolution to extract the channel features and reshape it back to its original size $C \times 2 \times 2$. Finally, apply UNAP to obtain the channel attention tensor $X_c$, which is multiplied by the initial input X to get the input $X_p^{in}$ for positional attention module. In summary, channel attention can be summarized as follows (Eq. (3)):

$$X_p^{in} = X \otimes UNAP\big(\sigma\big(C1D\big(X_c^{add}\big)\big)\big).  \tag{3}$$

Here, C1D represents 1D convolution, and σ denotes the sigmoid activation function.

Positional attention module: Fig. 6 shows the Positional Attention Module of FEHA. As demonstrated in CBAM, using the two modules sequentially proves more effective. Therefore, in the spatial attention module, the output feature tensor $X_p^{in}$ from the channel attention module serves as the input for the positional attention module. We start by reshaping the inputs, splitting both X and $X_p^{in}$ along the channel dimension into two halves, followed by a hybrid concatenation to form $X_{p1}$ and $X_{p2}$.

This concatenation with the original input more precisely locates the features within the spatial context of the input image. Strip pooling is then applied to $X_{c1}$ and $X_{c2}$ along the horizontal and vertical directions, respectively, to obtain the spatial positional attention coordinates. These coordinates are concatenated, and $1 \times 1$ convolution is used to fuse the features, preliminarily forming the positional attention features.

$$X_p^{strip} = C2D\big(Cat\big(SPool_h\big(X_{p1}\big), SPool_w\big(X_{p2}\big)\big)\big).  \tag{4}$$

Here, C2D represents 2D convolution, Cat stands for the concatenation operation, and $SPool_h$ and $Spool_w$ respectively represent strip pooling on the height dimension and strip pooling on the width dimension.

These are then split along the dimension used for concatenation and expanded to form directional positional attention tensors. These tensors are multiplied by $X_{p1}$ and $X_{p2}$ respectively to derive positional feature tensors $X_p^{add}$.

$$X_p^{add} = \left(X_{p1} \otimes e\left(X_{p1}^{strip}\right)\right) \oplus \left(X_{p2} \otimes e\left(X_{p2}^{strip}\right)\right). \tag{5}$$

Here, e represents the expand operation, and $X_{p1}^{strip}$ and $X_{p2}^{strip}$ respectively refer to the two tensors obtained by splitting $X_p^{strip}$ along the concatenated dimension.

The final convolution is used to merge the features, culminating in the positional attention tensor $X_p$. The initial input is multiplied by the newly obtained positional attention tensor to produce a new attention feature map, which is the final output of FEHA, as shown in Eq. (6).

$$X_p = X \otimes \sigma\left(C2D\left(X_p^{add}\right)\right). \tag{6}$$

## Sec-YOLO structure

In the field of unsafe behavior recognition, the primary focus is on detecting behaviors within surveillance video. Surveillance videos typically contain targets of varying scales, necessitating that network models possess excellent multi-scale fusion capabilities. Even within the same category of unsafe behavior, the characteristics can vary significantly, such as in the case of falling states. Falling represents a sudden target state, which can result in various falling forms, hence the model needs to effectively adapt to these changes in target types during feature extraction. Although YOLOv8n employs the C2f module and path aggregation network—feature pyramid network (PAN-FPN) (*Liu et al., 2018*; *Lin et al., 2017*) to enhance its feature extraction and multi-scale feature fusion capabilities, it still faces challenges in detecting unsafe behaviors with frequently changing scales and forms. To address this, we improved YOLOv8n by integrating various enhancement modules and incorporating the newly proposed FEHA attention mechanism. This led to the development of Sec-YOLO, a new model specifically designed for detecting common unsafe behaviors, as shown in Fig. 7.

## Spatial feature enhancement *via* RFAConv

Traditional convolutional operations apply the same set of weights across all spatial positions within the receptive field. This parameter sharing mechanism, while computationally efficient, limits the network's ability to capture the varying importance of different spatial features—particularly in complex scenes such as industrial environments where contextual cues may differ significantly across locations. Moreover, standard convolution lacks the capacity to emphasize semantically important regions within the receptive field, leading to reduced representational power. To address this issue, attention mechanisms were introduced and have been integrated into various networks to help the model concentrate on more informative features. Spatial attention, in particular, partially mitigates the rigidity of parameter sharing by adaptively weighting spatial positions. However, when used with large kernel sizes, conventional spatial attention still struggles to resolve the limitations of weight uniformity, as it often lacks the granularity to distinguish subtle yet critical spatial dependencies—especially in high-resolution surveillance imagery. RFAConv addresses these challenges by introducing receptive-field-aware spatial weighting. It enhances traditional convolution through a dual-branch design—one

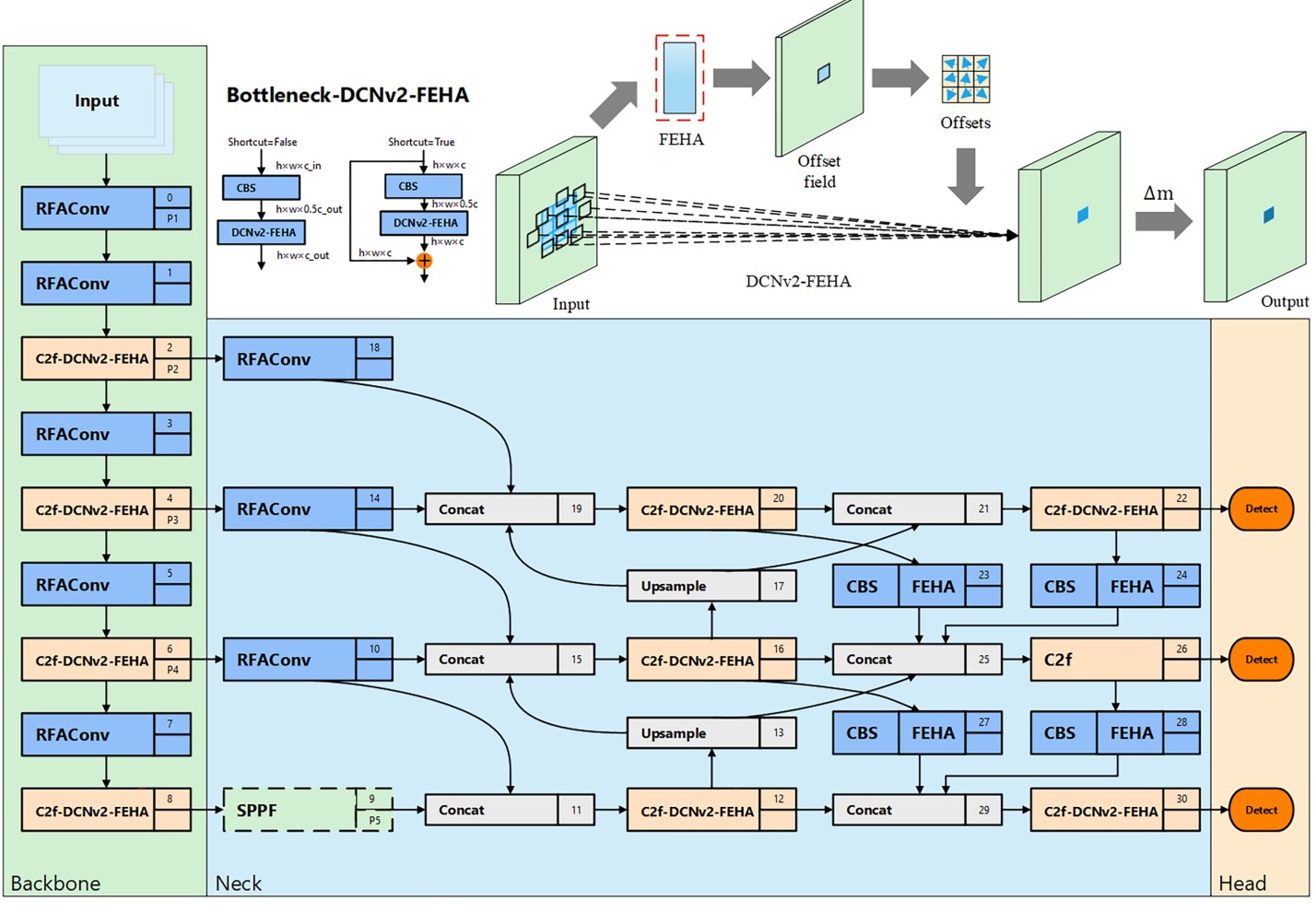

**Figure 7** Overall architecture of Sec-YOLO and the structure and principle diagram of Bottleneck-DCNv2-FEHA.

capturing local features *via* group convolution, and the other aggregating global context through average pooling. A SoftMax attention mechanism is then used to assign dynamic importance weights across the receptive field, allowing the model to focus more on semantically meaningful regions.

In the context of unsafe behavior detection, such as identifying whether a worker is falling or using a phone, the ability to discriminate between foreground action and background clutter is crucial. RFAConv allows the model to dynamically prioritize regions with high behavioral saliency—such as body posture, hand movement, or helmet visibility—while suppressing irrelevant background signals. This targeted enhancement significantly improves detection robustness in industrial scenes characterized by occlusion, scale variation, and cluttered visual information.

In terms of computational complexity, the introduction of RFAConv does not significantly increase the overall model size or FLOPs. Since it replaces the original convolution with lightweight group convolution and local attention mechanisms, the

overall structural footprint remains nearly unchanged. This ensures that improvements in spatial feature sensitivity are achieved without adding notable computational burden.

### Deformable feature extraction *via* DCNv2-FEHA

Standard convolution operations adopt fixed and uniform sampling patterns, which limits their ability to model spatially variant features in detection scenarios. This limitation is particularly problematic in industrial environments, where unsafe behaviors such as falling or sleeping on duty often involve significant variations in posture and motion trajectories. These spatial transformations make it difficult for traditional convolutional kernels to capture all contextually relevant features, leading to performance degradation in complex or dynamic scenes. To address these challenges, various convolutional variants have been proposed. Deformable convolutional networks (DCN) introduced learnable spatial offsets, enabling convolutional kernels to adaptively shift their sampling locations. The second-generation DCNv2 further incorporates modulation scalars to control the influence of each sampling point. While DCNv2 improves geometric flexibility, its offset prediction remains entirely based on local convolutional features, lacking semantic guidance. As a result, it may still focus on irrelevant regions under conditions such as occlusion, clutter, or low contrast. Other convolutional innovations have also attempted to handle spatial diversity. For instance, dynamic convolution (DyConv) dynamically selects convolutional kernels based on input content, enhancing feature adaptiveness—but at the cost of increased memory usage and inference complexity, which limits its applicability in real-time systems (*Chen et al., 2020*). Large kernel convolutions, as employed in architectures like RepLKNet (*Ding et al., 2022*) and ConvNeXt (*Liu et al., 2022*), can effectively capture long-range dependencies, but their high computational cost and lack of fine-grained local deformation modeling make them unsuitable for lightweight, real-time unsafe behavior detection.

Given these limitations, we propose an enhanced solution: DCNv2-FEHA, which integrates our FEHA mechanism directly into the offset and modulation prediction pathway of DCNv2. We choose DCNv2 over other deformable convolution variants due to its lightweight design and stable detection performance. By incorporating both global and local attention across channel and spatial dimensions, FEHA provides semantic guidance that steers the learned offsets toward behavior-relevant regions—such as the head, limbs, or hands. This guided deformation mechanism enhances the model's robustness in handling complex or occluded behaviors.

The resulting C2f-DCNv2-FEHA module is embedded into the YOLOv8 backbone by replacing the second convolution layer in the Bottleneck block. This integration significantly improves the network's ability to localize and recognize unsafe behaviors under diverse spatial conditions, without introducing substantial computational overhead—thus maintaining real-time performance.

Although DCNv2 introduces additional operations for offset learning and modulation, and FEHA adds attention refinement, their integration into the backbone is carefully controlled. The deformable convolution is selectively applied, and the FEHA module is designed with lightweight pooling and convolutional components. As a result, this

combined module provides enhanced spatial adaptability with only a minimal increase in model complexity.

## Multi-scale feature fusion *via* MAFPN-FEHA

To address the challenge of multi-scale feature fusion, we restructured the model's neck based on the MAFPN design. The novel neck efficiently integrates multi-level contextual information, enhancing the model's ability to detect objects across different scales. This improvement is particularly evident in complex scenes, where the model demonstrates superior performance.

Additionally, we introduced the FEHA attention mechanism into the deeper layers of the neck to further refine the fusion of contextual features, resulting in more precise and robust feature representations. This deep integration improves the model's performance on multi-scale features, effectively overcoming the challenges posed by variations in scale and morphology in unsafe behavior detection.

While MAFPN modifies the neck structure by introducing multi-branch fusion paths, including the superficial assisted fusion (SAF) and Advanced Assisted Fusion (AAF) modules, its overall computational complexity remains well-controlled. These modules operate on existing feature maps at predefined resolution levels and do not introduce additional deep branches or increase the spatial resolution of feature maps. Furthermore, the design emphasizes reusability and shallow-layer fusion rather than stacking or expansion, ensuring that additional operations are localized and efficient.

The integration of FEHA into the MAFPN structure, particularly when applied to deep layers, also follows a lightweight principle. FEHA itself relies on simple pooling and convolutional operations, and its insertion does not alter the overall architecture depth or introduce redundant computation paths. Even when FEHA is applied to both shallow and deep layers, the structural layout of the neck remains unchanged, and no new layers or channels are added.

As a result, although the MAFPN-FEHA combination enhances the model's ability to capture and integrate multi-scale features with attention guidance, the associated computational cost is marginal. The network remains within the computational range of a lightweight detector, making it suitable for deployment in environments with constrained hardware resources.

Through a series of targeted innovations, Sec-YOLO achieves significant improvements in the accuracy of unsafe behavior detection. The model demonstrates an enhanced ability to accurately capture the critical features of unsafe behaviors and exhibits exceptional adaptability to frequent morphological variations. Additionally, Sec-YOLO excels in identifying multi-scale unsafe behavior targets within complex scenarios. While there is a slight increase in parameters and computational cost, the increase is minimal, ensuring the model remains efficient even in resource-constrained environments. This balance of high performance and computational efficiency makes Sec-YOLO well-suited for a wide range of real-world applications.

## EXPERIMENT AND ANALYSIS

This section first introduces the dataset used, experimental parameters, and evaluation metrics. Subsequently, a series of experiments were conducted, including loss analysis of Sec-YOLO, hyperparameter selection, comparison with other feature fusion networks, and validation of the effectiveness of the proposed FEHA module. In addition, Sec-YOLO was compared with mainstream models, and ablation studies were performed to thoroughly analyze the effectiveness of the model improvements. Finally, the actual detection results of the model were presented to further demonstrate its superiority in unsafe behavior detection tasks.

### Datasets

In this experiment, the dataset employed was a custom collection compiled through online resources, designed to address the limitations of existing datasets for detecting unsafe behaviors. Current datasets typically focus on specific categories or include a variety of human actions without explicitly targeting multiple unsafe behaviors. To better represent diverse unsafe behaviors in industrial scenarios, we collected images from public datasets and publicly accessible websites, re-annotating them using the Make Sense annotation platform to create an integrated dataset. The dataset compiled and used in this study is available at DOI: 10.5281/zenodo.14015767.

The resulting dataset comprises 8,799 images representing four of the most common unsafe behaviors in industrial settings: falling (2,105 images), sleeping on duty (2,095 images), using mobile phones (2,200 images), and not wearing safety helmets (2,399 images). These images were divided into training, validation, and testing sets in a 7:2:1 ratio. To ensure representativeness and practical relevance, specific datasets were chosen based on their alignment with industrial unsafe behavior scenarios. For example, the UR Fall Detection Dataset (*Kwolek & Kepski, 2014*) and the Multicam Fall Dataset (*Auvinet et al., 2010*) were selected for their extensive coverage of falling incidents under varying conditions, supplemented by web searches to expand contextual diversity. Similarly, the Safety Helmet Wearing Dataset (SHWD) (*PaddlePaddle, 2022a*; *njvisionpower, 2019*) was included for its focus on key safety compliance behaviors in industrial settings. To address the scarcity of data for behaviors such as smartphone usage and sleeping on duty, additional efforts were made to enhance diversity. The smartphone usage dataset was obtained from publicly available dataset websites (*PaddlePaddle, 2022b*), while the sleeping-on-duty dataset was compiled by extracting frames from surveillance footage of workers and supplemented with web searches. This approach ensured the inclusion of realistic and diverse scenarios.

The data used in this study is publicly available and sourced from reputable datasets, which have been anonymized to protect individual privacy. Images from surveillance footage were selected and processed in accordance with data protection regulations, ensuring that no personally identifiable information was retained. In particular, the datasets utilized, including the SHWD, UR Fall Detection Dataset, and Multicam Fall Dataset, follow ethical guidelines for data collection and use, ensuring the privacy and confidentiality of individuals depicted. No personal or sensitive information was used in

the compilation of the dataset, and all annotations were performed in compliance with ethical standards. Furthermore, a double-review process was implemented to maintain annotation quality, ensuring consistency and accuracy across all categories.

To further improve the dataset's generalization capability, particular attention was paid to environmental diversity, such as variations in lighting conditions, camera angles, and worker demographics. Detailed annotation guidelines were established to ensure consistency during the annotation process. For falling behaviors, the falling states were annotated; for sleeping-on-duty behaviors, the sleeping postures of individuals at their workstations were labeled; for smartphone usage, the phones themselves were annotated; and for behaviors involving the absence of safety helmets, the head regions of targets were precisely labeled. To ensure the accuracy and reliability of annotations, a double-review process was employed, and random sampling was conducted for quality control.

## Experimental setup and evaluation metrics

This experiment was conducted on a Windows 10 operating system, utilizing the PyTorch deep learning framework. The experimental environment included Python 3.8 and CUDA 11.3. The hardware specifications comprised an 11th Gen Intel(R) Core(TM) i7-11700KF CPU at 3.60 GHz, 32 GB of RAM, and an NVIDIA GeForce RTX 3090 GPU with a boost clock of 1.70 GHz and 24 GB of VRAM. In the model training parameters, the learning rate was set to 0.01, with 200 training epochs, a batch size of 32, and the optimizer used was SGD.

In this experiment, to validate the advantages of the Sec-YOLO model, we employed several commonly used performance metrics, including precision, recall, mAP@0.5, mAP@0.5:0.95, and computational cost (GFLOPs). To properly understand the evaluation metrics used, we introduce definitions for four model detection outcomes: True Positive (TP), which represents the number of samples correctly predicted as positive by the model; False Positive (FP), which represents the number of samples incorrectly predicted as positive; True Negative (TN), which denotes the number of samples correctly predicted as negative; and False Negative (FN), which indicates the number of samples incorrectly predicted as negative.

Precision measures the proportion of actual positives among the samples predicted by the model as positive. It assesses the accuracy of positive predictions, that is, among all the samples predicted as positive, how many are truly positive. This can be expressed with Eq. (7):

$$Precision = \frac{TP}{TP + FP}. \tag{7}$$

Recall measures the proportion of actual positive samples that are correctly predicted as positive by the model. It evaluates the sensitivity of the model, indicating how many of the

actual positive samples are correctly identified as positive. This can be expressed with Eq. (8):

$$Recall = \frac{TP}{TP + FN}.$$ (8)

Mean average precision (mAP) is a widely used evaluation metric in object detection and information retrieval, utilized to assess the detection effectiveness of a model across all categories. It integrates both precision and recall to measure the overall performance of the model. To calculate mAP, one must first compute the average precision (AP), which is derived from the area under the precision-recall curve. A higher AP value indicates better model performance. mAP is the mean of the AP values across all categories and can be expressed by Eq. (9):

$$mAP = \frac{1}{N} \sum_{i=1}^{N} AP_i.$$ (9)

Here, $N$ represents the number of categories, and $AP_i$ is the average precision for the ith category. mAP@0.5 refers to the mAP calculated at an Intersection over Union (IoU) threshold of 0.5. IoU is a measure of overlap between the predicted bounding box and the true bounding box. A prediction is considered correct when the IoU reaches or exceeds 0.5. mAP@0.5:0.95 represents the average of the mAPs calculated at multiple IoU thresholds from 0.5 to 0.95, with an interval of 0.05. This means that the model's performance is evaluated at various IoU thresholds, providing a more comprehensive assessment of its effectiveness.

Billion floating-point operations per second (GFLOPs) is a key metric for measuring the computational complexity of neural network models, representing the number of billion floating-point operations that a model can perform per second. This metric is particularly important when assessing the performance of neural network models in processing scientific computations, graphics, and machine learning tasks. It is commonly used to describe the efficiency of a model running on graphics processing units (GPUs) or other high-performance computing systems. A lower GFLOPs value indicates that the model is computationally less demanding, making it more efficient and lightweight, suitable for deployment in resource-constrained environments.

Parameters (Params) is a key metric for measuring the complexity and capacity of neural network models. Fewer parameters result in a more streamlined model, lower risk of overfitting, and reduced demand on computational resources. This makes the model more suitable for deployment in resource-constrained environments such as mobile devices and embedded systems. In summary, the number of parameters directly relates to the model's efficiency and applicability, particularly in settings with limited resources, making a smaller number of parameters the preferable choice.

## Loss curve analysis

In our research, we conducted a thorough analysis of the loss curves of the Sec-YOLO model after training. As shown in Fig. 8, the training and validation losses of the model

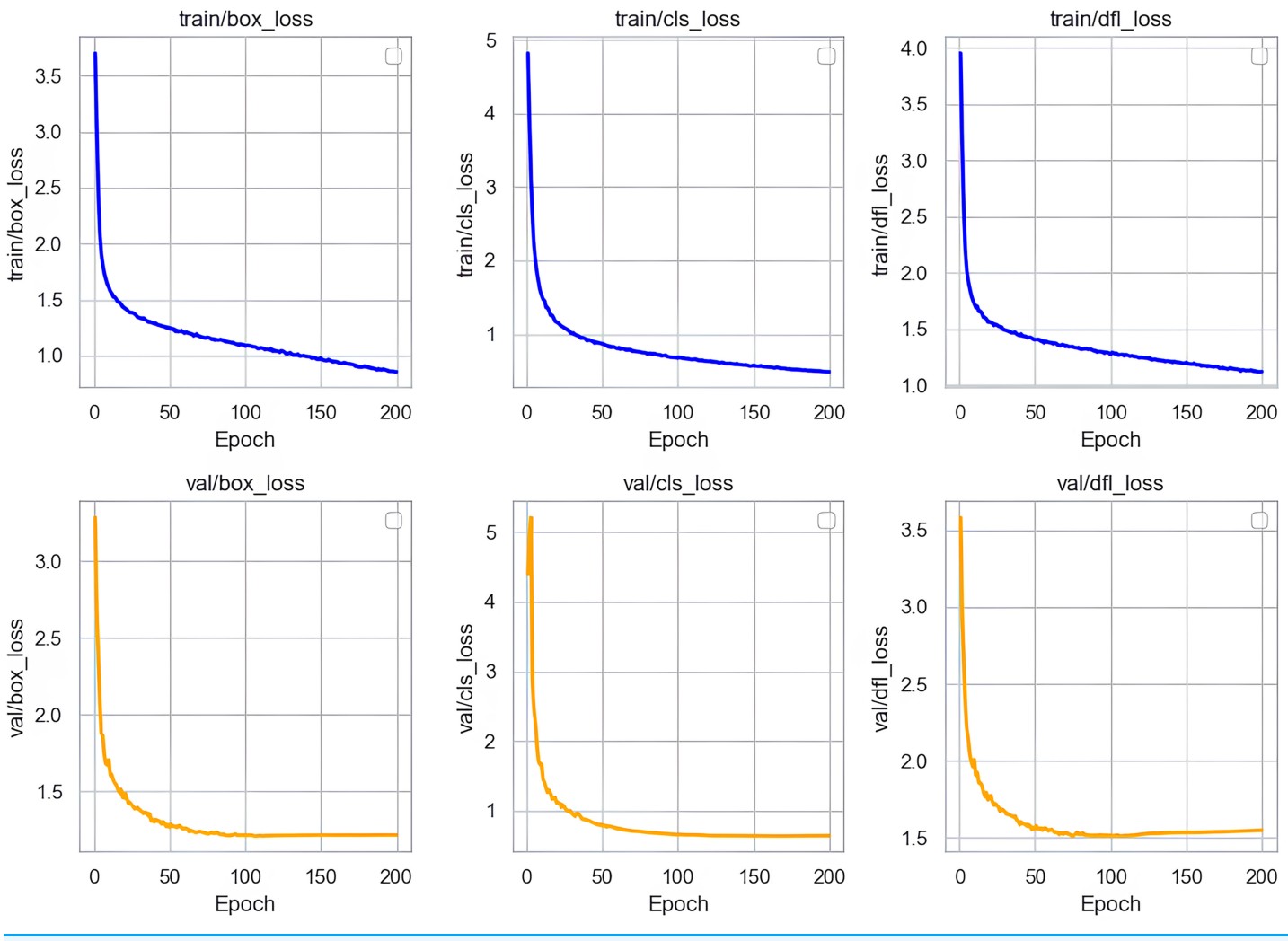

**Figure 8** **Training and validation losses of Sec-YOLO.**     

consist primarily of three components: box_loss, cls_loss, and dfl_loss. Box_loss assesses the accuracy of the model's predicted bounding box positions on the training set. Its loss value decreases gradually from an initially high level and stabilizes, indicating a progressive improvement in the model's ability to predict bounding box positions, which also demonstrates good generalization capabilities on unseen data. Cls_loss evaluates the accuracy of classifications on the training dataset, and the downward trend of this loss indicates that the model becomes increasingly precise in classifying objects, also showing good classification performance on the validation set.

Dfl_loss specifically addresses the common issue of class imbalance in object detection and focuses on enhancing the model's efficiency in handling small targets and difficult samples. During training, the rapid decrease and subsequent stabilization of dfl_loss indicate that the model effectively learned to balance differences between classes, improving recognition of small targets and difficult samples in complex scenes. Similarly, the continued reduction of dfl_loss on the validation set further validates the model's

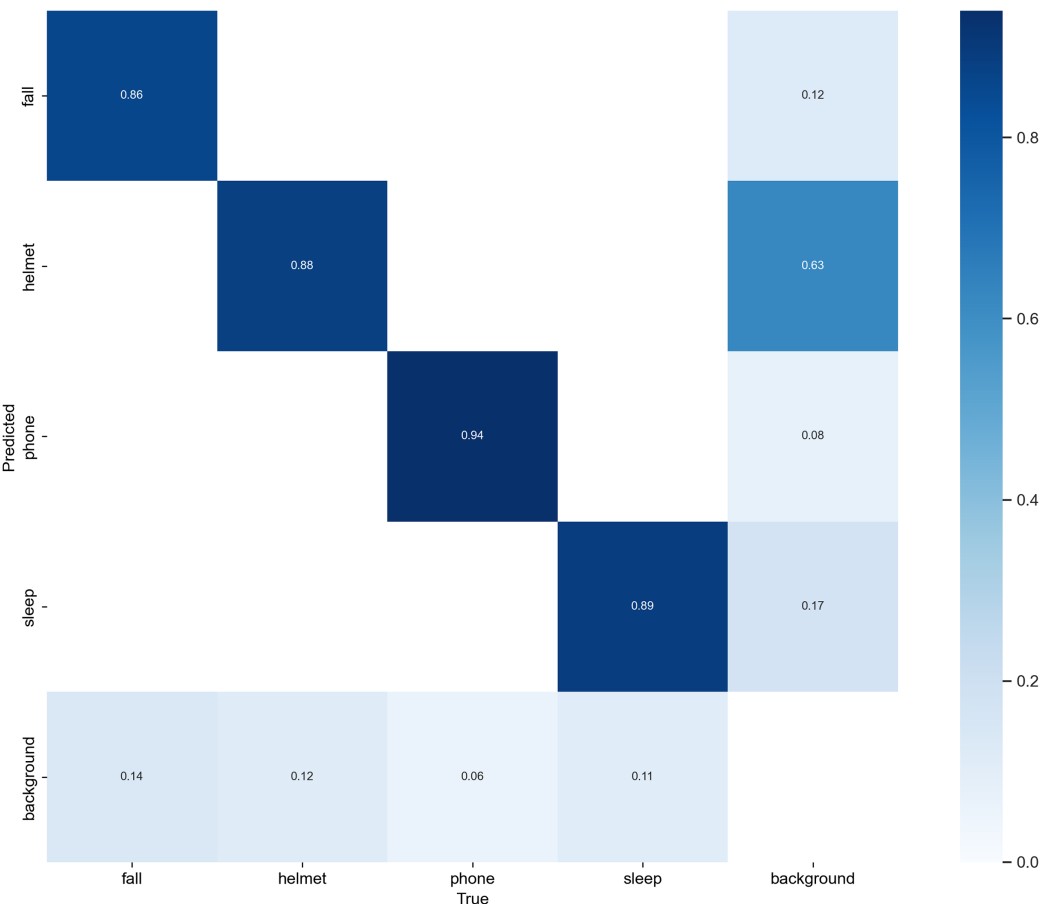

**Figure 9 Normalized confusion matrix of Sec-YOLO.** The horizontal axis represents the ground truth categories, while the vertical axis shows the predicted categories. Each cell displays the proportion of predictions for a given true class, with rows normalized to sum to 1. Higher diagonal values indicate correct predictions, while off-diagonal values represent confusion between classes. The matrix reflects that Sec-YOLO achieves high class-wise accuracy and effective discrimination in unsafe behavior detection tasks.

ability to process new data, effectively resolving issues related to class imbalance and the challenges of recognizing small targets and difficult samples.

In summary, the analysis of these loss curves demonstrates that the Sec-YOLO model exhibits rapid convergence and high stability during training and validation processes. The model's strong performance and feasibility in practical applications are evident from its good generalization on the validation dataset.

## Confusion matrix analysis

To comprehensively assess the behavior-wise classification performance of Sec-YOLO, we visualized the normalized confusion matrix on the custom dataset, as illustrated in Fig. 9. This analysis complements the quantitative metrics such as mAP by revealing the inter-class prediction dynamics and the model's discriminative capability for each behavior category.

As shown in the matrix, Sec-YOLO achieves high classification consistency across all major unsafe behavior categories, with true positive rates of 94% for phone, 89% for sleep, 88% for helmet, and 86% for fall. These results demonstrate that the architectural components of Sec-YOLO—such as FEHA attention modules and multi-scale enhancements—contribute effectively to capturing behavior-specific visual features under various scenarios. The model shows strong reliability in differentiating both static and dynamic postures, even under challenging conditions like occlusions or motion blur.

While the majority of predictions are highly accurate, a small portion of background samples were predicted as unsafe behaviors such as fall or helmet. Similarly, a subset of fall instances were predicted as background. These observations are common in behavior detection tasks, where ambiguous poses or overlapping features may exist across categories, especially in industrial environments with diverse camera angles and complex backgrounds.

Importantly, these cases are relatively infrequent and do not significantly affect the model's overall detection accuracy. On the contrary, the low confusion rates affirm that the model generalizes well across categories while maintaining a high degree of category separation. Moreover, these insights can inform future enhancements such as incorporating richer contextual cues or multi-frame temporal consistency, especially in real-time monitoring applications.

In conclusion, the confusion matrix supports the effectiveness of the Sec-YOLO architecture in multi-class unsafe behavior detection, and reflects its robustness in distinguishing complex postures with minimal class overlap, thereby validating the practical viability of the proposed system.

## Hyperparameter selection

In the channel attention component of FEHA, we utilize a "convolution + pooling" method to compensate for the errors that may arise from the combination of average pooling and max pooling. Additionally, we introduce a hyperparameter $\alpha$ to adjust the magnitude of this compensation. While $\alpha$ is set to be learnable, its initial value has a significant impact on the model's early-stage training, thereby influencing overall model performance. To explore this effect, we conducted experiments with initial values of $\alpha$ ranging from 0 to 0.9, with a step size of 0.1 (where 0 indicates no compensation). The results, shown in Fig. 10, indicate that an initial value of 0.5 provides the most optimal performance.

When the alpha is set to 0, the model's mAP is 92.1%, relying entirely on the unprocessed fusion of average and max pooling results, unaffected by the "convolution + pooling" compensation. However, as alpha increases to 0.5, the mAP rises to its peak at 92.6%, demonstrating that the balanced integration of "convolution + pooling" compensation with the original pooling methods offers the best feature representation for the current dataset. Further increasing alpha to 0.9 leads to a decrease in mAP to 91.9%, indicating that the model increasingly depends on the compensated results, which can sometimes result in overemphasis on certain features while neglecting other critical ones. This trend suggests that introducing "convolution + pooling" compensation generally

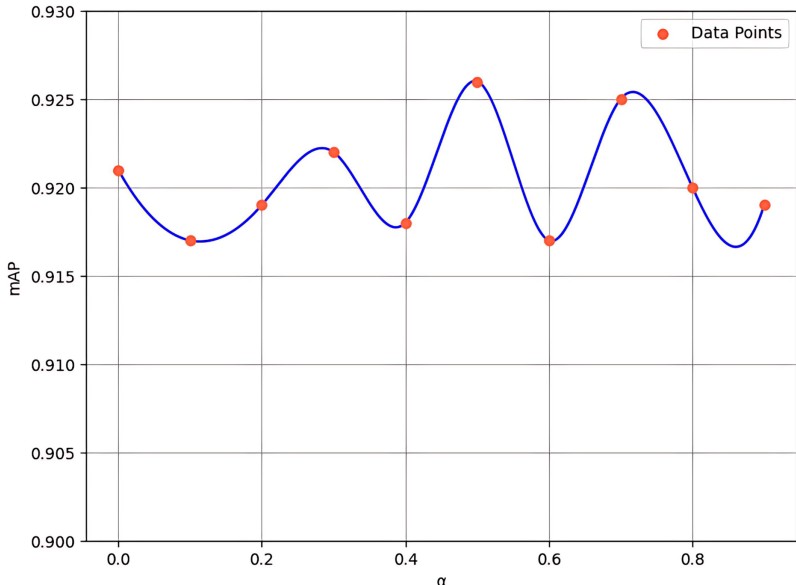

**Figure 10 Experimental results for various values of hyperparameter α.**

benefits the model's performance as alpha increases from 0 to 0.5. However, when the alpha value is excessively high, the model may become overly sensitive to noise or insignificant features. Thus, excessive reliance on processed features can compromise the overall performance of the model.

Experimental results indicate that 0.5 serves as an optimal balance point. Although the "convolution + pooling" approach is introduced as a compensation mechanism, it effectively extracts meaningful features, which may lead to improved performance.

## Compare with other feature fusion networks

In our experiments, we utilized the backbone network of YOLOv8n and conducted multiple replacements of its neck component to test the effects of different network architectures: First, we replaced the neck part with the classical BiFPN (*Tan, Pang & Le, 2020*); then, we used the content-guided attention fusion (CGAFusion) from DEA-Net (*Chen, He & Lu, 2024*); followed by the attentional scale sequence fusion (ASF) from ASF-YOLO (*Kang et al., 2024*); in addition, based on HS-FPN from MFDS-DETR (*Chen et al., 2024*), we also implemented a replacement and, inspired by the path aggregation network (PAN), designed and included HS-PAN in our experiments; finally, the neck part was replaced with MAFPN from the model used in this article, MAF-YOLO.

The experimental results are displayed in the Table 1, where MAFPN demonstrated the most outstanding performance, achieving an mAP@0.5:0.95 of 61.4%, with a reduction in parameter count. Although BiFPN, HS-FPN, and HS-PAN reduced the computational and parameter requirements by an average of 1.0 GFLOPs and 1.1 M respectively, their detection accuracy significantly decreased, which is unacceptable in unsafe behavior detection tasks. For CGA Fusion, while its computational load and parameter count

**Table 1 Comparison results with other feature fusion networks.** The experiment utilizes four evaluation metrics: mAP@0.5/%, mAP@0.5 :0.95/%, GFLOPs, and the number of parameters (Params, in millions). These metrics were selected because they provide a direct illustration of the performance and complexity of different feature fusion networks.

| Module | mAP@0.5/% | mAP@0.5:0.95/% | GFLOPs | Params (M) |
|---|---|---|---|---|
| Baseline | 90.6 | 61.1 | 8.7 | 3.2 |
| BiFPN | 90.2 | 60.7 | 7.7 | 2.1 |
| CGA fusion | 90.7 | 60.8 | 9.0 | 3.3 |
| ASF | 91.0 | 61.0 | 9.4 | 3.3 |
| HS-FPN | 90.3 | 59.6 | 7.5 | 2.0 |
| HS-PAN | 90.5 | 59.7 | 7.7 | 2.2 |
| MAFPN | 90.8 | 61.4 | 9.4 | 3.1 |

increased relative to YOLOv8n, similar to MAFPN, its mAP was not as high as that of MAFPN. The closest in performance to MAFPN was ASF, which even exceeded MAFPN by 0.2% in mAP@0.5; however, MAFPN had a smaller parameter count and outperformed ASF by 0.4% in mAP@0.5:0.95, making it more suitable for unsafe behavior detection tasks in terms of overall capability.

## Compare with attention mechanisms

To evaluate the effectiveness of the proposed FEHA attention mechanism, we conducted comparative experiments with various attention mechanisms. Initially, we incorporated the squeeze-and-excitation (SE) attention mechanism (*Hu, Shen & Sun, 2018*), a well-established approach widely recognized for enhancing the channel attention capabilities of models. Subsequently, we compared FEHA with other mainstream attention mechanisms, including CBAM and HAM, which enhance features by combining channel attention with spatial or positional attention. Additionally, we included the CA attention mechanism in our comparison due to its outstanding performance in positional information extraction. Finally, to explore the effectiveness of different mechanisms in efficient local and global information interaction, we introduced the lightweight attention mechanism MLCA into our experiments. The experimental results are shown in Table 2.

As our attention mechanism is embedded within deformable convolutions, the validation experiment to assess FEHA's effectiveness in enhancing the model's ability to extract key features and spatially locate critical regions was conducted using YOLOv8n with C2f-DCNv2 as the baseline network. In this experiment, five different attention mechanisms were embedded in the deformable convolutions for comparative analysis, while all other settings remained unchanged. Through a comprehensive analysis of the results, it was found that FEHA achieved superior composite scores, with an mAP@0.5 of 92.1%, representing increases of 0.7% over the baseline network.

After integrating the SE module, the model's recall rate significantly improved, reaching the highest value in this experiment at 86.4%. However, its accuracy was unsatisfactory, at only 89.8%, which directly led to a decrease in mAP. This issue may arise because the SE module primarily focuses on the interdependencies between channels, potentially

**Table 2 Comparative results with various attention mechanisms.** In this experiment, the attention mechanisms embedded into DCNv2 have a minimal impact on the model's parameter count and computational load. Consequently, the focus of this study is primarily on the performance improvements brought about by these attention mechanisms. To this end, the selected evaluation metrics include Precision/%, Recall/%, mAP@0.5/%, and mAP@0.5:0.95/%.

| Module | Precision/% | Recall/% | mAP@0.5/% | mAP@0.5:0.95/% |
|---|---|---|---|---|
| Baseline | 91.6 | 84.2 | 91.4 | 62.2 |
| +SE | 89.8 | 86.4 | 91.1 | 61.7 |
| +CBAM | 92.2 | 84.5 | 91.9 | 62.5 |
| +CA | 91.3 | 84.6 | 91.5 | 62.3 |
| +MLCA | 91.3 | 84.6 | 91.1 | 62.3 |
| +HAM | 90.5 | 84.5 | 91.1 | 61.0 |
| +FEHA | 92.2 | 84.9 | 92.1 | 62.5 |

overlooking other critical feature dimensions, especially spatial features. While the recall rate increased, the rise in false positives resulted in lower overall accuracy, which negatively impacted the model's overall performance.

In contrast, the integration of the CBAM module led to a substantial improvement in the model's overall performance compared to the baseline network. CBAM combines both channel and spatial attention mechanisms, allowing the model to capture features more accurately. This combination enhances the network's attention to multi-dimensional features, effectively mitigating the potential biases that arise when focusing on only one aspect, thus improving both accuracy and mAP.

The HAM module, which integrates channel and spatial position features, was designed with a focus on lightweight architecture. However, this resulted in a trade-off where some accuracy was sacrificed for model simplicity. Consequently, HAM did not perform as well as expected in this experiment. Although its structure is more streamlined, its ability to handle complex features was diminished, leading to performance that fell short of the other attention mechanisms.

The CA module, which emphasizes spatial position features, also contributed to a performance increase. By enhancing spatial information capture, the CA module improved the model's recognition capability, though the extent of the improvement was relatively modest.

The MLCA module, designed as a lightweight attention mechanism that focuses on both local and global feature interactions, did not demonstrate a significant enhancement in overall model performance in this experiment. Despite its lightweight nature, MLCA failed to exhibit a substantial improvement in the model's results.

Finally, the proposed FEHA module outperformed the baseline network across all evaluation metrics, demonstrating exceptional performance. The FEHA module effectively combines the advantages of various attention mechanisms, ensuring comprehensive attention to both channel and spatial features while striking a balance between model complexity and accuracy. Experimental results show that FEHA achieved the best scores in terms of accuracy, mAP@0.5, and mAP@0.5:0.95, reaching 92.2%, 92.1%, and 62.5%,

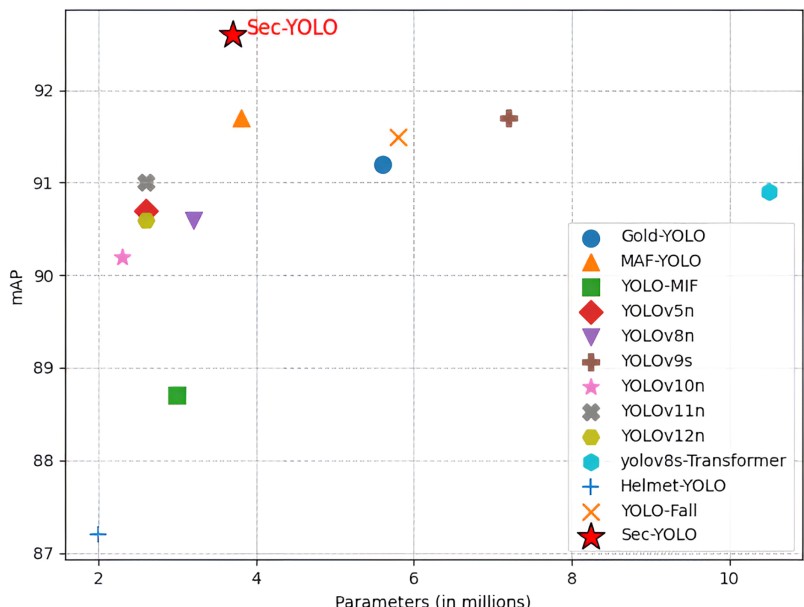

**Figure 11 Comparison results with the current model.**

respectively. These results confirm the superiority of the FEHA module in handling complex tasks.

## Comparison with the current models

To comprehensively assess both the computational efficiency and detection accuracy of the proposed Sec-YOLO model, we conducted extensive comparative experiments against a range of mainstream YOLO models and recent task-specific variants. Given the importance of real-time deployment and resource constraints in industrial settings, the comparison focuses not only on mAP performance but also on computational complexity metrics such as GFLOPs and parameter count. The selected models span multiple generations of the You Only Live Once (YOLO) family, including YOLOv5n, YOLOv8n, YOLOv9s (*Wang, Yeh & Liao, 2024*), YOLOv10n (*Wang et al., 2024*), YOLOv11n (*Khanam & Hussain, 2024*), and YOLOv12n (*Tian, Ye & Doermann, 2025*), as well as customized variants like GOLD-YOLO (*Wang et al., 2023*), MAF-YOLO, and YOLO-MIF (*Wan et al., 2024*), and domain-specific detectors such as yolov8s-Transformer (*Do et al., 2023*), Helmet-YOLO (*Zhou et al., 2024*), and YOLO-Fall (*Zhao et al., 2024*).

As shown in Fig. 11, Sec-YOLO achieves the highest mAP@0.5 of 92.6%, with only 3.7 M parameters and 9.6 GFLOPs, outperforming all comparison models in accuracy while maintaining moderate complexity. For instance, YOLOv9s, which shares the same mAP@0.5 of 91.7%, requires almost double the computation (26.7 GFLOPs) and significantly more parameters (7.2 M). Lighter models such as YOLOv10n (2.3 M, 6.7 GFLOPs) and YOLOv5n (2.6 M, 7.7 GFLOPs) offer lower computational load but suffer from reduced accuracy, reaching only 90.2% and 90.7% mAP, respectively.

When compared with task-specific detectors, Sec-YOLO maintains its advantage in both accuracy and complexity. While MAF-YOLO achieved the same 91.7% accuracy, it requires more computation (10.5 GFLOPs). GOLD-YOLO (91.2%) and YOLO-Fall (91.5%) both fall short in performance, despite having larger model sizes and higher GFLOPs (12.1 and 12.3, respectively).

In terms of extreme configurations, Helmet-YOLO (2.0 M, 6.5 GFLOPs) is the lightest model but yields only 87.2% accuracy, while yolov8s-Transformer, the heaviest model (10.5 M, 29.2 GFLOPs), achieves only 90.9%, confirming that increased complexity does not always lead to better results.

These findings highlight that Sec-YOLO achieves an optimal trade-off between accuracy and computational cost, making it suitable for real-world deployment in industrial safety applications where both detection reliability and resource constraints must be balanced.

## Ablation experiments

In the Sec-YOLO model, we introduced RFAConv to replace the traditional CBS module, which not only enhances the model's focus on receptive field spatial features but also significantly improves feature extraction efficiency. Additionally, this study proposed the FEHA attention mechanism, which was integrated into DCNv2, enhancing the offset capturing ability of DCNv2. This led to the formation of the C2f-DCNv2-FEHA structure, replacing the original C2f and markedly enhancing the model's ability to extract features from irregular variations. In the neck section, inspired by MAFPN, we reconstructed the original PAN-FPN architecture of YOLOv8n. Experimental results showed that adding the FEHA attention mechanism to the four CBS units in the deep layer of the reconstructed neck structure effectively enhances the model's ability to integrate multi-scale features.

To verify the actual impact of these improvements on model performance, we conducted a series of ablation studies. Considering the effectiveness of FEHA in optimizing DCNv2 and MAFPN, we first verified the enhancements in these two parts. Finally, we integrated the three improved components—RFAConv, C2f-DCNv2-FEHA, and MAFPN-FEHA—into a comprehensive ablation experiment, comparing the effectiveness of each improvement step by step.

### *DCNv2 optimized with FEHA*

This section demonstrates the improvement effect of DCNv2 on the model and further validates the optimization impact of FEHA on DCNv2. As shown in Table 3, after integrating DCNv2 into the C2f module, the overall computational load of the model is reduced compared to the original YOLOv8n, despite DCNv2 involving additional offset and modulation computations. This reduction is largely attributed to the improved feature representation capability of DCNv2, which enables the model to achieve better accuracy with fewer redundant operations.

Specifically, the model achieves increases of 1.0% and 0.9% in mAP@0.5 and mAP@0.5:0.95, respectively. When FEHA is further incorporated into DCNv2 to guide the offset generation, performance is enhanced across all evaluation metrics—reaching 93.0% in mAP@0.5—while maintaining a compact computational profile. These results

**Table 3 Performance of DCNv2 optimized with FEHA.**

| Module | Precision/% | Recall/% | mAP@0.5/% | mAP@0.5:0.95/% |
|---|---|---|---|---|
| YOLOv8n | 90.3 | 83.1 | 90.6 | 61.1 |
| +DCNv2 | 91.6 | 84.2 | 91.4 | 62.1 |
| +DCNv2-FEHA | 92.2 | 84.9 | 92.1 | 62.5 |

**Table 4 Performance of MAFPN optimized with FEHA.**

| Module | Precision/% | Recall/% | mAP@0.5/% | mAP@0.5:0.95/% |
|---|---|---|---|---|
| YOLOv8n | 90.3 | 83.1 | 90.6 | 61.1 |
| +MAFPN | 91.8 | 82.9 | 90.8 | 61.4 |
| +MAFPN-Shallow | 90.9 | 84.3 | 90.8 | 61.0 |
| +MAFPN-All | 92.5 | 83.0 | 90.8 | 61.2 |
| +MAFPN-Deep | 90.8 | 85.5 | 91.5 | 61.9 |

demonstrate that the combination of DCNv2 and FEHA not only improves detection performance but also preserves model efficiency.

### MAFPN optimized with FEHA

This section demonstrates the improvement effects of MAFPN on the model, further substantiates the efficacy of FEHA in enhancing feature fusion within MAFPN, and identifies the optimal placement of FEHA within the MAFPN structure. According to Table 4, the introduction of MAFPN into the neck results in enhanced performance metrics across various aspects compared to the original YOLOv8n model. We integrated FEHA at the shallow layers of the neck, specifically post-convolutions labeled as 10, 14, and 18 in Fig. 8, denoted in Table 4 as "+MAFPN-Shallow." FEHA was also added to the deeper layers, post-convolutions 23, 24, 27, and 28, represented in Table 4 as "+MAFPN-Deep." Additionally, FEHA was applied to both shallow and deep layers, indicated as "+MAFPN-All" in the Table 4.

As shown in Table 4, after integrating the FEHA module into the shallow layers, the model's performance declined compared to the model with only the MAFPN module. Although there was a significant improvement in recall, accuracy decreased, and mAP@0.5:0.95 dropped by 0.4%. This is because when FEHA is integrated into the shallow layers, which are in the early stages of feature fusion, the model primarily performs initial fusion of features from different layers. Introducing FEHA at this stage may cause the model to overly emphasize certain features, neglecting others that are also important. As a result, this can negatively impact the model's generalization ability, ultimately leading to a decrease in accuracy. When the FEHA module was added to both the shallow and deep layers of MAFPN, the model did not exhibit a substantial improvement compared to the model with only MAFPN. While accuracy increased by 0.7%, there was no improvement in mAP@0.5, and mAP@0.5:0.95 decreased by 0.2%. This suggests that placing the attention mechanism in both shallow and deep layers may lead to redundant or conflicting attention signals. The shallow layers could provide too much localized attention to less informative

features, while the deep layers may not have had enough focus on high-level abstractions due to the interference from the shallow attention mechanism. This misalignment in attention flow might hinder the model's ability to extract and fuse meaningful features across different scales. However, when the FEHA module was integrated into the deep layers of MAFPN, the model showed improvements across various evaluation metrics. Compared to the model with only MAFPN, despite a slight drop in precision, there was a significant increase in recall and mAP values, with improvements of 2.6%, 0.7%, and 0.5%, respectively. This suggests that integrating the FEHA module into the deep feature extraction layers has a positive impact on enhancing the overall performance of the model. In deep layers, features are more abstract and contain high-level representations of the input data. At this stage, applying attention mechanisms can help the model focus on the most relevant, semantically rich features, which are crucial for improving the overall performance of the model. The increase in recall indicates that the attention mechanism helped to capture more relevant positive samples, while the improvement in mAP highlights better feature discrimination for precise localization and recognition tasks.

We did not include GFLOPs and parameter statistics in Table 4, as all the FEHA integration strategies—whether applied to shallow layers, deep layers, or both—result in the same overall model complexity. Specifically, each configuration maintains approximately 3.4 million parameters and 9.5 GFLOPs, with no measurable variation across strategies. This is because the placement of FEHA within MAFPN only alters the location of attention enhancement, not the number of layers or operations, as the module is reused in identical form and number.

### Comparison of improvements in each part

Following the discussions above, the effectiveness of FEHA in optimizing DCNv2 and MAFPN has been demonstrated. Therefore, the next section will further validate the impact of these optimizations through stepwise combination ablation experiments on the RFAConv, DCNv2-FEHA, and MAFPN-FEHA optimization modules. The related experimental results are displayed in Table 5.

As shown in Table 5, Model 1 represents the baseline YOLOv8n, while Models 2, 3, and 4 correspond to the individual application of RFAConv, DCNv2-FEHA, and MAFPN-FEHA, respectively. Each of these modifications contributes positively to model performance, with Model 3 showing the most significant improvement. Not only does it achieve notable increases in precision and recall, but it also improves the key metrics mAP@0.5 and mAP@0.5:0.95 by 1.7% and 2.1%, respectively, over Model 1. Additionally, the GFLOPs decrease from 8.7 to 8.5, indicating that DCNv2-FEHA plays a crucial role in feature extraction within the backbone network. Models 4 through 7 demonstrate the effects of combining these modifications in pairs. It is observed that, although the combined use of RFAConv and DCNv2-FEHA results in a slight decrease in mAP@0.5 compared to DCNv2-FEHA alone, other performance metrics show improvements, with only minimal increases in GFLOPs and Params. The combination of RFAConv and MAFPN-FEHA also proves effective, while the pairing of DCNv2-FEHA and MAFPN-FEHA yields enhancements across all metrics, particularly with a 3.2% increase

**Table 5 Results of ablation studies.**

| Model | RFAConv | DCNV2+FEHA | MAFPN+FEHA | P/% | R/% | mAP@0.5/% | mAP@0.5:0.95/% | GFLOPs | Params (M) |
|-------|---------|------------|------------|-----|-----|-----------|----------------|--------|------------|
| 1 | | | | 90.3 | 83.1 | 90.6 | 61.1 | 8.7 | 3.2 |
| 2 | ✓ | | | 90.8 | 83.9 | 91.6 | 61.5 | 9.1 | 3.2 |
| 3 | | ✓ | | 92.2 | 84.9 | 92.1 | 62.5 | 8.5 | 3.4 |
| 4 | | | ✓ | 90.8 | 85.5 | 91.5 | 61.9 | 9.5 | 3.4 |
| 5 | ✓ | ✓ | | 91.8 | 85.2 | 92.0 | 63.0 | 8.8 | 3.4 |
| 6 | ✓ | | ✓ | 91.7 | 84.1 | 91.4 | 62.1 | 9.8 | 3.5 |
| 7 | | ✓ | ✓ | 91.6 | 85.3 | 92.2 | 63.3 | 9.2 | 3.6 |
| 8 | ✓ | ✓ | ✓ | 91.3 | 86.5 | 92.6 | 63.6 | 9.6 | 3.7 |

**Note:**
"P" stands for Precision, "R" represents Recall, and "√" indicates that the corresponding improvement module has been integrated into the model.

in mAP@0.5:0.95. Finally, Model 8, which integrates all three improvements—referred to as Sec-YOLO—demonstrates the best performance in both mAP@0.5 and mAP@0.5:0.95.

In addition to performance improvement, we also examined the computational impact of each module introduced in Sec-YOLO. As shown in Table 5, the inclusion of RFAConv, DCNv2-FEHA, and MAFPN-FEHA leads to marginal increases in GFLOPs and parameter count compared to the YOLOv8n baseline. Specifically, the full Sec-YOLO model has 3.7 M parameters and 9.6 GFLOPs, which remains in the lightweight category and is comparable to YOLOv8n (3.2 M, 8.7 GFLOPs). This demonstrates that our modular enhancements are computationally efficient and do not compromise the model's suitability for real-time deployment.

## Detection effect analysis

We conducted detection experiments in real-world scenarios, selecting a set of challenging images that included all categories present in the dataset. Detection was performed on these images using both YOLOv8n and Sec-YOLO, and the results were compared. Furthermore, to more intuitively observe Sec-YOLO's attention to features, we generated heatmaps that provide a clearer visualization of the model's feature-capturing capabilities.

To ensure ethical compliance and subject anonymity, all images used for visualization and heatmap analysis were anonymized prior to publication. Human regions were manually annotated and masked in gray, while regions corresponding to specific behavior categories were masked in blue. All non-annotated background areas were blurred using strong Gaussian filtering. Heatmaps were overlaid as transparent pseudo-color layers to visualize model attention without revealing any sensitive information.

Figure 12 illustrates the performance differences between YOLOv8n and Sec-YOLO in complex multi-scale object detection scenarios. YOLOv8n exhibits significant deficiencies, particularly in detecting small-scale objects, frequently resulting in both false positives and missed detections. Detailed observations include: an erroneously marked safety helmet in the first image; a non-existent falling event incorrectly identified in the second image; the third image shows an overall better performance, yet fails to detect a small-scale safety helmet in the distance; the fourth image, despite successfully identifying all targets, suffers

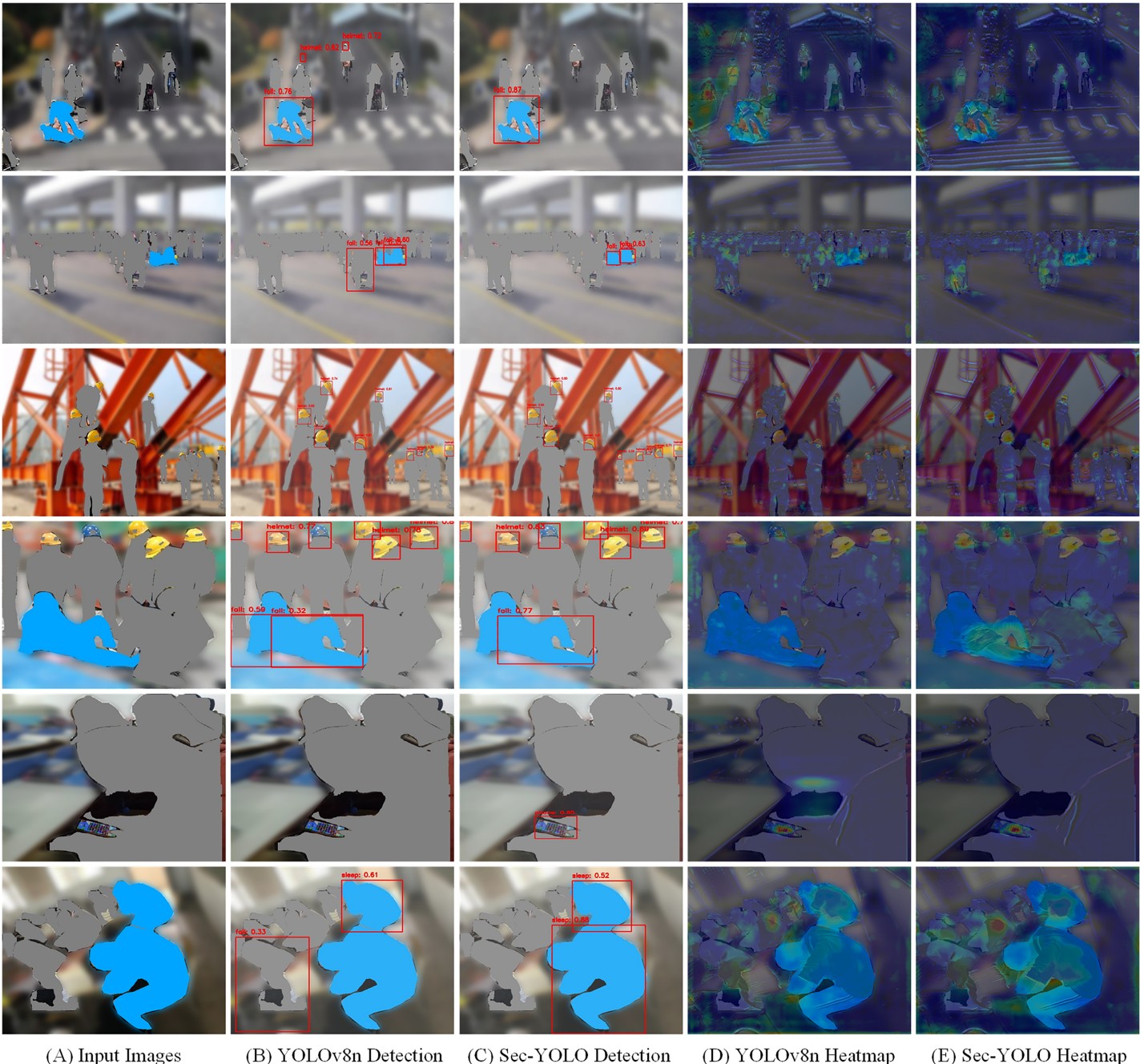

|                      |                       |                       |                     |                    |
| -------------------- | --------------------- | --------------------- | ------------------- | ------------------ |
| (A) Input Images     | (B) YOLOv8n Detection | (C) Sec-YOLO Detection | (D) YOLOv8n Heatmap | (E) Sec-YOLO Heatmap |

**Figure 12 Comparative detection results and heatmap analysis between Sec-YOLO and YOLOv8n.** (A) Input image, (B) YOLOv8n results, (C) Sec-YOLO results, (D) YOLOv8n heatmap, (E) Sec-YOLO heatmap. Human regions are anonymized in gray, with blue masks indicating specific behaviors. Backgrounds are blurred for privacy.

from overlapping detection boxes; the fifth image misses an existing mobile phone; and the sixth image not only falsely reports a falling event but also misses a sleeping at work incident.

In contrast, Sec-YOLO demonstrates exceptional detection accuracy across all test images, with no occurrences of false positives or missed detections. It shows superior scale adaptability, accurately identifying targets of dramatically different scales, as seen in the fourth image. Furthermore, in detecting small objects, as in the third and fifth images, Sec-YOLO exhibits high precision and meticulous target recognition capabilities.

The heat map analysis of the models reveals significant disparities in feature attention. YOLOv8n, despite recognizing key features of targets, often misdirects its attention to non-target areas, particularly evident in the first and fifth images, leading to a high rate of false positives. This scattered attention pattern may stem from the model's excessive sensitivity to environmental noise. Conversely, the heat maps of Sec-YOLO display concentrated and precise feature recognition, sharply focusing on target features, which aids in maintaining consistent detection performance across various complex backgrounds and effectively avoiding false recognition of non-target areas. These results distinctly showcase the fundamental differences in design and optimization strategies between the two models in object detection technology.

To better demonstrate the performance of Sec-YOLO compared to the latest YOLO series models, we incorporate YOLOv9s, YOLOv10n, YOLOv11n, and the latest YOLOv12n in the experiments, and generate heatmaps for visualization. The experimental results are shown in Fig. 13. Specifically, in the fall detection task, although all five models can capture the target features, Sec-YOLO exhibits more precise delineation of the target contours. In helmet detection, YOLOv11n and YOLOv12n also show strong feature capturing capabilities, but Sec-YOLO maintains a more balanced feature extraction for each target. In mobile phone usage detection, all five models perform well, but Sec-YOLO is more accurate in capturing the contours of the target. In nap detection, Sec-YOLO stands out particularly, as it not only successfully captures the features of the two targets but also significantly reduces the risk of false positives that other models may produce. Overall, Sec-YOLO demonstrates superior performance compared to the latest YOLO series models, showcasing its distinct advantage in unsafe behavior detection.

## Generalization test on public datasets

To evaluate the generalization capability of Sec-YOLO and rule out the possibility of overfitting to the custom dataset, we conducted additional experiments on two publicly available datasets: the Safety Helmet Wearing Dataset (SHWD) and the CAUCA Fall Dataset (*Eraso et al., 2022*). These datasets contain the same unsafe behavior categories—not wearing a helmet and falling—as our custom dataset, but differ significantly in terms of scene layouts, camera angles, body postures, lighting conditions, and visual domains. The results are summarized in Table 6.

The SHWD dataset primarily focuses on helmet-wearing compliance in industrial environments, with diverse backgrounds, lighting conditions, and human postures. To ensure a fair evaluation, we excluded all SHWD images used in the training phase of our

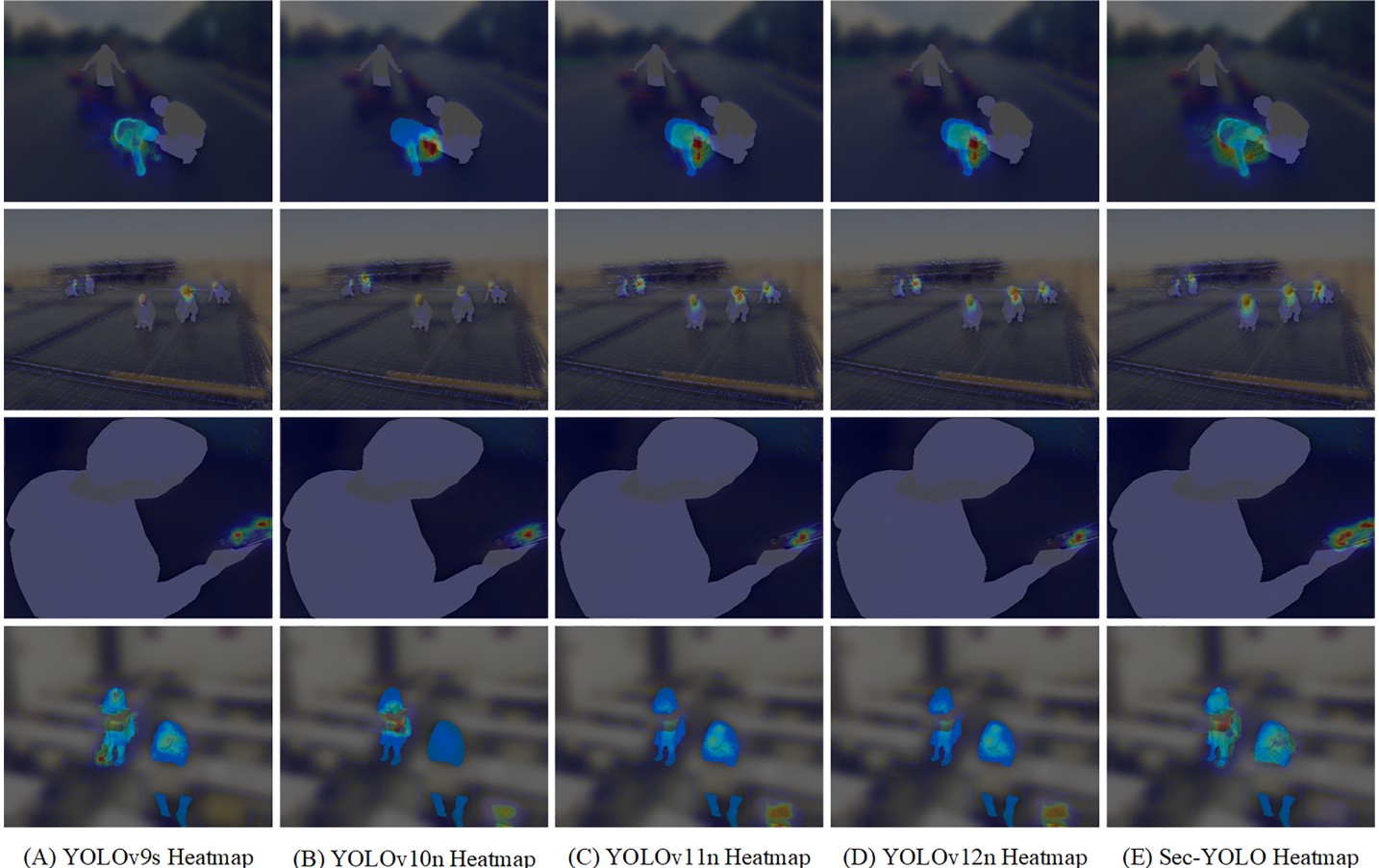

(A) YOLOv9s Heatmap     (B) YOLOv10n Heatmap     (C) YOLOv11n Heatmap     (D) YOLOv12n Heatmap     (E) Sec-YOLO Heatmap

**Figure 13 Heatmap comparing Sec-YOLO with the YOLO series.** (A) YOLOv9s, (B) YOLOv10n, (C) YOLOv11n, (D) YOLOv12n, (E) Sec-YOLO. All human regions were anonymized by masking in gray, with blue masks indicating behavior-specific annotations. Backgrounds were blurred to protect contextual privacy.

**Table 6 Performance of Sec-YOLO on custom and public datasets for generalization evaluation.**

| Dataset | Precision/% | Recall/% | mAP@0.5/% | mAP@0.5:0.95/% |
|---|---|---|---|---|
| Custom dataset | 91.3 | 86.5 | 92.6 | 63.6 |
| SHWD | 90.1 | 83.6 | 89.8 | 61.1 |
| CAUCA fall | 88.4 | 80.9 | 87.3 | 58.7 |

custom dataset. This eliminated any potential bias due to data leakage or memorization. We tested the pre-trained Sec-YOLO model on the remaining subset of SHWD without further fine-tuning. The model achieved 89.8% mAP@0.5 on this subset, demonstrating strong generalization to industrial safety behaviors in unfamiliar environments.

The CAUCA Fall Dataset is a benchmark for fall detection, containing various human fall postures from different directions. To help the model better recognize fall characteristics, the dataset includes images of visually similar postures such as sitting, kneeling, and picking up objects. We applied the trained Sec-YOLO model to this dataset,

achieving 87.3% mAP@0.5, indicating that the model successfully transferred its learned representations of unsafe behaviors to a different set of postures and scenarios.

Although the unsafe behavior categories in the public datasets align with those in our custom dataset, the significant differences in visual style, body posture distribution, and environmental context make them ideal benchmarks for evaluating the model's generalization ability. Sec-YOLO's strong performance on both datasets validates its ability to detect unsafe behaviors not only within the original training domain but also across various visual conditions, highlighting its practical applicability in real-world safety monitoring systems.

## CONCLUSIONS

This study proposes Sec-YOLO, a novel target detection model for identifying unsafe behaviors. The model employs the RFAConv module in the backbone network to improve feature representation for critical target areas. To handle morphological diversity, the DCNv2 module is incorporated into C2f structure, improving the model's generalization and robustness to diverse target shapes. To further improve detection performance in complex multi-target and multi-scale scenarios, the neck section is reconstructed based on the MAFPN structure, which effectively integrates shallow and deep features. Additionally, by integrating the novel FEHA mechanism into the DCNv2 module, the precision of offsets and feature convolution accuracy are significantly improved, resulting in enhanced detection performance. These improvements enable the model to more accurately capture the critical features of various unsafe behaviors while adapting to their morphological variations. The model's performance in multi-scale target detection has been significantly enhanced, particularly in identifying small targets in industrial scenarios, which increases its reliability and practicality in real-world applications. Experimental evidence demonstrates that integrating FEHA into the deep feature fusion section of MAFPN significantly optimizes feature fusion and extraction. On a custom unsafe behavior dataset, Sec-YOLO has shown superior detection performance, achieving an mAP@0.5 of 92.6% and an mAP@0.5:0.95 of 63.6%, confirming its high accuracy in feature capture in practical applications. Future work will focus on expanding the dataset to include more types of unsafe behavior data, optimizing the algorithm, and exploring integration with edge computing devices for real-time detection. Additionally, temporal models, such as LSTM, will be considered for future video-based detection, where capturing temporal dependencies across frames could further improve the detection of dynamic behaviors.

### Funding

This work was supported by the S&T Program of Hebei (NO. 22370701D). The funders had no role in study design, data collection and analysis, decision to publish, or preparation of the manuscript.

## Grant Disclosures

The following grant information was disclosed by the authors:
S&T Program of Hebei: 22370701D.

## Competing Interests

The authors declare that they have no competing interests.

## Author Contributions

- Yang Liu conceived and designed the experiments, performed the experiments, analyzed the data, performed the computation work, prepared figures and/or tables, and approved the final draft.
- Shuaixian Liu performed the experiments, analyzed the data, performed the computation work, prepared figures and/or tables, and approved the final draft.
- Jie Gao conceived and designed the experiments, analyzed the data, authored or reviewed drafts of the article, and approved the final draft.
- Tao Song conceived and designed the experiments, performed the computation work, authored or reviewed drafts of the article, and approved the final draft.
- Wenyu Dong performed the experiments, prepared figures and/or tables, and approved the final draft.

## Data Availability

The code and the dataset are available at Zenodo

- zrgh111. (2024). zrgh111/Sec-YOLO: Sec-YOLO (Sec-YOLO). Zenodo. https://doi.org/10.5281/zenodo.14233626.

- liu, yang. (2024). Insecure datasets [Data set]. Zenodo. https://doi.org/10.5281/zenodo.14015768.

## Supplemental Information

Supplemental information for this article can be found online at http://dx.doi.org/10.7717/peerj-cs.3151#supplemental-information.

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
