# Peer review of "Detection of unsafe workplace behaviors: Sec-YOLO model with FEHA attention"

_PeerJ Computer Science, doi:10.7717/peerj-cs.3151_

## Round 0.1 · original submission · Minor Revisions

Thank you for submitting your manuscript to PeerJ Computer Science. After careful review, the reviewers have raised some concerns regarding the methodology and experimentation that need to be addressed before we can proceed with the publication.

We kindly request that you revise your manuscript in light of the reviewers' comments and make the necessary adjustments. Please also provide a detailed response letter addressing each of the reviewers' suggestions and observations.

We are confident that, with these revisions, your manuscript will be considered for publication.

Thank you again for your contribution, and we look forward to receiving your revised submission.

**PeerJ Staff Note**: Please ensure that all review, editorial, and staff comments are addressed in a response letter and that any edits or clarifications mentioned in the letter are also inserted into the revised manuscript where appropriate.

Reviewer 1 ·

Basic reporting

Review

1. I believe that this study will be useful in terms of literature in terms of its subject. I congratulate the authors.
2. This Study has been reviewed by other referees before me.
3. I consider the recommendations made by the reviewers for the study to be correct and appropriate.
4. The authors have implemented the recommendations in their studies. Thank you for that.
5. The study is considered successful in its general structure and as a result of the changes made.
6. The study is more detailed and proper in terms of literature and model explanation.


The study is considered adequate in terms of basic reporting. I have a few suggestions about the validity of the findings. I believe that if these suggestions are realized, it will be more successful academically.

Experimental design

The use of the customized dataset in the study demonstrated the success of the study in terms of the model. The parameters used were chosen correctly.
There is no ethical problem in the study.

Validity of the findings

Suggestions
I believe that the study is suitable for publication in a journal in this form. However, it can be enriched a little more academically.
1. The experimental study can be diversified due to the use of a generalized dataset in the study. This will reveal whether there is memorization in the YOLO architecture.

2. Even if the study was considered adequate in terms of experimental studies, the study could have been reanalyzed with an LSTM-style model.

good studies.

Additional comments

The study is considered successful in terms of its general structure as it has been examined before and the authors have taken the suggestions into consideration.

In this form, the study is suitable for publication in a journal. I have a few suggestions for the validity of the findings. If these suggestions are fulfilled, academic validity and reliability will be increased.

In general, the study successfully explained the problem and the solution. Literature research is sufficient. The results are presented in proper detail. Presentation of experimental outputs is adequate. Confusion-Matrix table can also be added.

Good work.

Cite this review as

Reviewer 2 ·

Basic reporting

In this paper, a model called Sec-YOLO is proposed to detect unsafe human behavior. The model is compared with object detection algorithms that have been successful in solving the problems in which they have been used. The results obtained are also given in tables. The study contributes to the literature in terms of the research topic and the results obtained. When the study is evaluated in terms of content and form, it is recommended to be accepted in this form.

Experimental design

no comment

Validity of the findings

no comment

Additional comments

no comment

Cite this review as

·

Basic reporting

The content of the article, together with the consistent and accurate use of technical terms, demonstrates the ability to clearly explain complex concepts. The information provided is clear and technically correct. The language of the article has a professional and understandable style.
The technical explanations of the article (details and operations of modules such as RFAConv, DCNv2, MAFPN, FEHA) are appropriate and correct to the current information in the field of deep learning. The overall tone of the text is objective and professional, which adheres to the standards of courtesy and expression expected in scholarly publishing.
The article provides sufficient background and context on industrial safety and the role of human factors in production accidents. The "Introduction" and "RELATED STUDIES" sections highlight the limitations of current security management measures and the importance of intelligent electronic surveillance methods. Furthermore, the relevant previous literature, such as the YOLOv8n architecture, RFAConv, DCNv2, MAFPN, and various attention mechanisms, are properly cited.
The article clearly poses the problem of the detection of human behavior in the field of industrial safety, showing how the study fits into the wider field of knowledge. The inadequacies of traditional methods and the need for intelligent surveillance systems are explained in detail. Appropriate and extensive citations have been made to relevant previous studies (e.g., Faster R-CNN, LSTM-based approaches, YOLOv5 enhancements, and various attention mechanisms).
The article includes standard sections such as "Abstract", "Introduction", "Model Design Methodology", "Model Improvements", "Experimental Analysis Results and Discussions" and "Conclusion". This is an acceptable and professional article structure
Various figures are referred to in the text, from "Figure 1" to "Figure 12", and tables from "Table 1" to "Table 5". Loss curves of these figures and tables, hyperparameter analysis, comparisons with different networks, ablation studies and detection results were visualized,
The article is publicly available at DOI: 10.5281/zenodo.14015767 of the dataset used. This indicates that the raw data (or processed data set) is being made available in accordance with data sharing policies.
The article contains all the basic components of a research article (abstract, introduction, methodology, experiments, conclusion and citations) and represents a self-sufficient "publication unit".
The article exposes the limitations of current models and the hypothesis that Sec-YOLO will overcome these limitations and provide improvements in the detection of unsafe behavior. The experimental results directly support these hypotheses, with Sec-YOLO showing significant improvements in mAP values over YOLOv8n (92.6% [email protected]% and 63.6% [email protected]:0.95%) and outperforming other state-of-the-art models.
The article focuses on a specific problem, such as the detection of unsafe human behavior in industrial environments. The proposed Sec-YOLO model integrates interrelated and synergistic architectural innovations such as RFAConv, DCNv2-FEHA, and MAFPN-FEHA. Extensive experimental validation and ablation studies indicate that these innovations are covered in one coherent study. This suggests that the study was not inappropriately subdivided with the aim of increasing the number of publications, but rather represented a holistic research effort.
The article clearly defines all the key terms, including the key components of Sec-YOLO (RFAConv, DCNv2, MAFPN, FEHA) and evaluation metrics (Sensitivity, Recall, mAP, GFLOPs, Parameters). In the field of deep learning, "detailed proofs" are often presented in the form of mathematical formulations and empirical experimental results. The paper provides mathematical formulations for the channel attention module of RFAConv and FEHA and presents extensive experimental results, ablation studies, and comparative analyses to support the model's effectiveness.

Experimental design

The article addresses areas such as industrial security, computer vision, deep learning and artificial intelligence applications. Therefore, it is considered to be fully compatible with the purpose and scope of the journal. The study directly contributes to the accumulation of knowledge in this area by addressing the inadequacies of existing security management measures.
The article defines the research question very clearly: "Detecting unsafe human behaviors is crucial for improving safety in industrial production environments." and "Current models face limitations in multiscale target detection in such environments." It has been made clear that the main purpose of the study is to eliminate the inadequacies of existing detection systems in order to prevent accidents caused by human factors in industrial safety.
The introduction highlights the urgency and importance of the issue by presenting statistical data that the vast majority of industrial accidents (up to 73%) are caused by human factors. This reveals that the research question is both highly relevant and socially and industrially significant.
The article notes that current intelligent tracking methods face challenges such as "high target density, scene occlusion, and multi-scale detection issues." Furthermore, Faster R-CNN details the shortcomings of previous studies such as LSTM-based approaches and YOLOv5, such as slow detection speed, limited generalizability, or high computational costs. It is emphasized that Sec-YOLO fills this gap as a mission-oriented development of the YOLOv8n model, specially adapted to the complexities of industrial scenarios. The experimental results show that Sec-YOLO fills this gap, increasing detection accuracy and maintaining efficiency.
The study used a custom dataset of 8,799 images that included four common unsafe behaviors. The data collection process (online sources, publicly available datasets, surveillance footage), data split ratio (7:2:1), and rigorous annotation guidelines (double review, random sampling) are described in detail. The DOI of the dataset is also provided.
The equipment and training parameters on which the experiments are carried out are clearly indicated.
Widely accepted metrics such as Sensitivity, Recall, [email protected], [email protected]:0.95, GFLOPs, and Parameters are used, and their definitions and formulas are presented.
The article conducted extensive ablation studies that systematically evaluated the individual and combined contributions of each proposed module (RFAConv, DCNv2-FEHA, MAFPN-FEHA). Furthermore, the comparison of the performance of Sec-YOLO with mission-oriented detectors with YOLOv8n and other state-of-the-art YOLO models demonstrates the technical rigor of the study.
Analysis of loss curves (box_loss, cls_loss, dfl_loss) shows the rapid convergence and stability of the model, while detailed ablation analysis for the alpha hyperparameter reveals how the optimal value is determined.
The use of heatmaps visually supports the model's feature learning and attention capabilities.
All in all, the article presents a technically quite rigorous investigation, and its methods are largely reproducible. However, although attention has been paid to the protection of personal data in the images used in the article, the dataset is ambiguous, and since it is a study involving human supervision, more ethical standards and confidentiality issues will increase the holistic quality of the article.

Validity of the findings

The results were presented using quantitative metrics (mAP scores, GFLOPs, parameters) and qualitative analyses (detection impact analysis, heat maps). Tables and figures are used to summarize the results of comparative and ablation studies, which increases the comprehensibility of the results.

The paper's original research question is to overcome the limitations of existing models in detecting unsafe human behavior in industrial settings. The results presented show that Sec-YOLO overcomes these limitations, demonstrating superior accuracy and efficiency compared to the base model YOLOv8n and other state-of-the-art models. Improvements in mAP and the model's ability to handle multiscale objects and disordered behavior are attributed to the problem statement.

The "Conclusion" section summarizes the findings and reiterates the model's improvements in feature representation, generalization to various target shapes, multiscale detection, and overall performance. It also outlined future research directions (further data collection, integration with edge computing devices).

---

## Round 0.2 · accepted · Accept

I hope this message finds you well. After carefully reviewing the revisions you have made in response to the reviewers' comments, I am pleased to inform you that your manuscript has been accepted for publication in PeerJ Computer Science.

Your efforts to address the reviewers’ suggestions have significantly improved the quality and clarity of the manuscript. The changes you implemented have successfully resolved the concerns raised, and the content now meets the high standards of the journal.

Thank you for your commitment to enhancing the paper. I look forward to seeing the final published version.

Reviewer 2 ·

Basic reporting

The article may be published in this form.

Experimental design

No comment

Validity of the findings

No comment

Additional comments

No comment

Cite this review as

·

Basic reporting

This study offers innovative and comprehensive approaches to the detection of potentially dangerous behaviors. The researchers’ command of the subject matter and the effort they have invested are evident in each chapter. In particular, the detailed analyses comparing different methods enhance the scientific depth of the study. The attention to personal data confidentiality and adherence to ethical standards are important factors that strengthen the article’s credibility. Overall, I believe the findings will make significant contributions to the scientific community and add value to the body of knowledge in this field. I support the acceptance of this article for publication.

Experimental design

This study offers innovative and comprehensive approaches to the detection of potentially dangerous behaviors. The researchers’ command of the subject matter and the effort they have invested are evident in each chapter. In particular, the detailed analyses comparing different methods enhance the scientific depth of the study. The attention to personal data confidentiality and adherence to ethical standards are important factors that strengthen the article’s credibility. Overall, I believe the findings will make significant contributions to the scientific community and add value to the body of knowledge in this field. I support the acceptance of this article for publication.

Validity of the findings

This study offers innovative and comprehensive approaches to the detection of potentially dangerous behaviors. The researchers’ command of the subject matter and the effort they have invested are evident in each chapter. In particular, the detailed analyses comparing different methods enhance the scientific depth of the study. The attention to personal data confidentiality and adherence to ethical standards are important factors that strengthen the article’s credibility. Overall, I believe the findings will make significant contributions to the scientific community and add value to the body of knowledge in this field. I support the acceptance of this article for publication.